# Control of protein palmitoylation by regulating substrate recruitment to a zDHHC-protein acyltransferase

Fiona Plain[1,4], Jacqueline Howie[2,4], Jennifer Kennedy[2], Elaine Brown[2], Michael J. Shattock [3], Niall J. Fraser [1✉] & William Fuller [2✉]

Although palmitoylation regulates numerous cellular processes, as yet efforts to manipulate this post-translational modification for therapeutic gain have proved unsuccessful. The Na-pump accessory sub-unit phospholemman (PLM) is palmitoylated by zDHHC5. Here, we show that PLM palmitoylation is facilitated by recruitment of the Na-pump α sub-unit to a specific site on zDHHC5 that contains a juxtamembrane amphipathic helix. Site-specific palmitoylation and GlcNAcylation of this helix increased binding between the Na-pump and zDHHC5, promoting PLM palmitoylation. In contrast, disruption of the zDHHC5-Na-pump interaction with a cell penetrating peptide reduced PLM palmitoylation. Our results suggest that by manipulating the recruitment of specific substrates to particular zDHHC-palmitoyl acyl transferases, the palmitoylation status of individual proteins can be selectively altered, thus opening the door to the development of molecular modulators of protein palmitoylation for the treatment of disease.

[1] School of Medicine, University of Dundee, Dundee, UK. [2] Institute of Cardiovascular and Medical Sciences, College of Medical, Veterinary and Life Sciences, University of Glasgow, Glasgow, UK. [3] Cardiovascular Division, The Rayne Institute, King's College London, London, UK. [4] These authors contributed equally: Fiona Plain, Jacqueline Howie. ✉email: n.fraser@dundee.ac.uk; will.fuller@glasgow.ac.uk

Palmitoylation is the reversible attachment of the fatty acid palmitate to cysteine thiols via a thioester bond[1]. Many different classes of protein are palmitoylated, including G-proteins[2–4], ion channels, transporters and their regulators[5–7], receptors[8], as well as kinases[9,10]. Palmitoylation induces substantial changes in the secondary structure and therefore function of intracellular regions of target proteins through their recruitment to the surface of a membrane bilayer via the palmitoylated cysteine. Since palmitoylation dynamically and directly regulates the localization, interactions, turnover, and activity of numerous classes of protein[7,11–14], and can have a profound impact on the landscape of the plasma membrane[15,16], manipulation of protein palmitoylation with, for example, small molecules has the potential to exert substantial changes to cellular function.

Protein palmitoylation is catalyzed by a family of zinc finger and DHHC motif containing palmitoyl acyltransferase enzymes (zDHHC-PATs), reversed by protein thioesterases, and occurs dynamically and reversibly throughout the secretory pathway in a manner analogous to phosphorylation[17]. zDHHC-PATs are characterized by a cysteine-rich region with a conserved Asp-His-His-Cys (DHHC) motif[17] within the active site[18]; there are 23 human isoforms. They typically have four transmembrane (TM) domains, with a conserved ca. 50 amino acid cytosolic core between TM2 and 3 containing the zDHHC motif. In contrast, the intracellular amino and carboxyl termini are poorly conserved, and contribute to zDHHC isoform substrate selectivity[17]. The acyltransferase zDHHC5 is primarily located at the cell surface[19] where it regulates a diverse range of physiological processes. Known zDHHC5 substrates include somatostatin receptor 5[20], sphingosine-1-phosphate receptor 1[21], the lipid raft protein flotillin 2[22], the Na-pump regulatory protein phospholemman (PLM)[7,13], desmoglein-2 and plakophilin-3[23] as well as the neuronal proteins δ-catenin[24] and GRIP 1[25].

Unlike kinases, for which there are an ever-increasing number of inhibitors approved for use in the clinic[26,27], no therapeutically useful molecular modulators of protein palmitoylation have been developed thus far. This, in no small part, is due to a fundamental lack of knowledge regarding the molecular basis of both enzyme catalysis and substrate recruitment by zDHHC-PATs. To date, efforts to discover inhibitors of zDHHC-PATs have largely focussed on the enzyme's active site targeting catalytic activity (e.g.,[28–30]). However, the palmitoylation status of specific proteins may also be changed by selectively manipulating their recruitment to (a) cognate zDHHC-PAT(s).

The ubiquitous Na-K ATPase (Na-pump) is a P-type ATPase and is located at the surface of eukaryotic cells where it carries out the active export of 3 sodium (Na) and import of 2 potassium (K) ions per catalytic cycle across the plasma membrane. The Na pump is a multisubunit enzyme complex comprising an α (catalytic core of the enzyme that contains the Na, K and ATP binding sites), a β (required for trafficking of the Na pump through the secretory pathway) and a FXYD (regulates the kinetic properties and/or substrate affinities of the Na-pump) sub-unit[31]. PLM (FXYD1) regulates the Na-pump in muscle by way of post-translational modifications to its cytosolic C tail. PLM phosphorylation activates and PLM palmitoylation inhibits the Na-pump[31,32]. The Na-pump is not the only P-type ATPase regulated by a small palmitoylated accessory sub-unit. The SERCA 2a regulators phospholamban[33] and sarcolipin[34] are both palmitoylated within their transmembrane domains suggesting that palmitoylation may play an important general role in regulating P-type ATPase activity.

Previously, we have shown in cardiac muscle that the Na-pump interacts with zDHHC5 in a region immediately following transmembrane domain 4 (αα 218–334) and that this contact is required for palmitoylation of PLM[13]. Here, we show that PLM palmitoylation can be altered by manipulating the interaction between zDHHC5 and the Na-pump α sub-unit via post-translational modifications (palmitoylation, GlcNacylation) to zDHHC5 as well as with a pharmacological tool. In short, we show that the palmitoylation status of a specific substrate protein can be modified through controlling its interaction with a partner zDHHC-PAT.

## Results

**Mapping the Na-pump binding site on zDHHC5.** Previously, we have shown by truncation analysis that a region on zDHHC5 located between N218 and T334 is required for Na-pump binding and PLM palmitoylation[13]. The position of the binding site of the Na-pump on zDHHC5 was determined more exactly here by purifying the Na-pump from detergent-solubilized rat ventricular homogenates with an array of overlapping biotinylated 25mer peptides that covered the acyltransferase region αα 218–334. Western Blot analysis revealed a Na-pump binding peak comprising three overlapping peptides with the αα 233–257 peptide (termed Na-pump binding site peptide (NBSP)) displaying highest levels of pulldown (Fig. 1a). To gain insights into the tertiary position of the Na-pump binding site on zDHHC5, the primary sequence of the zDHHC5 C-terminal domain was compared with that of zDHHC20 whose crystal structure has recently been reported[35] aligning the sequences at the highly conserved TTxE motif (Fig. 1b). zDHHC5 and 20 have appreciable homology (40% identity, 73% similarity) across their palmitoyltransferase conserved C-terminus (PaCCT) motifs[36] suggesting that they likely have similar structures in this region. The zDHHC20 PaCCT motif contains an amphipathic helix (αα 264–271)[35] which is conserved in zDHHC5 (αα 237–244) also (Fig. 1c). The NBSP contains the amphipathic helix in its entirety suggesting that this structural feature may be important for binding between zDHHC5 and the Na-pump. The amphipathic helix is flanked at its N-terminus by two cysteine residues (C236, C237) (Fig. 1d), both of which have previously been shown to be palmitoylated[37]. The proximity of these modifiable residues to the Na-pump binding site on zDHHC5 suggested to us that Na-pump recruitment to zDHHC5 may be regulated by post-translational modification. Surprisingly, the position of the Na-pump binding site on zDHHC5 was to be found a considerable distance from the enzyme active site (Fig. 1e), suggesting that this region of the acyltransferase does not interact directly with PLM.

**zDHHC5 interacts with the Na-pump α-sub-unit.** The interaction between zDHHC5 and the Na-pump occurs in the cytosol[13]. To address whether or not zDHHC5 interacts with PLM directly or via one of the other (α, β) pump subunits, a series of biotinylated peptides covering the PLM intracellular C-tail were tested for their ability to purify HA-tagged zDHHC5 from HEK cells. zDHHC5 could not be purified with any of the peptides including one comprising the entire PLM C-tail (Fig. 2a). However, the PLM peptides all purified the Na-pump α subunit as expected demonstrating that the pulldown experiment had been successful (Fig. 2a). Together, these results confirm that PLM and zDHHC5 do not interact directly.

To establish which Na-pump sub-unit zDHHC5 interacts with, the NBSP was used to purify the Na-pump from whole rat heart lysates either as an intact complex or as individual subunits following dissociation of the pump using harsh detergents. Whole rat heart homogenate was lysed with four different detergents (C12E10, Triton, SDS, and FC-12) and each lysate divided in two. Biotinylated NBSP was added to one half of each lysate and binding partners captured with streptavidin sepharose beads.

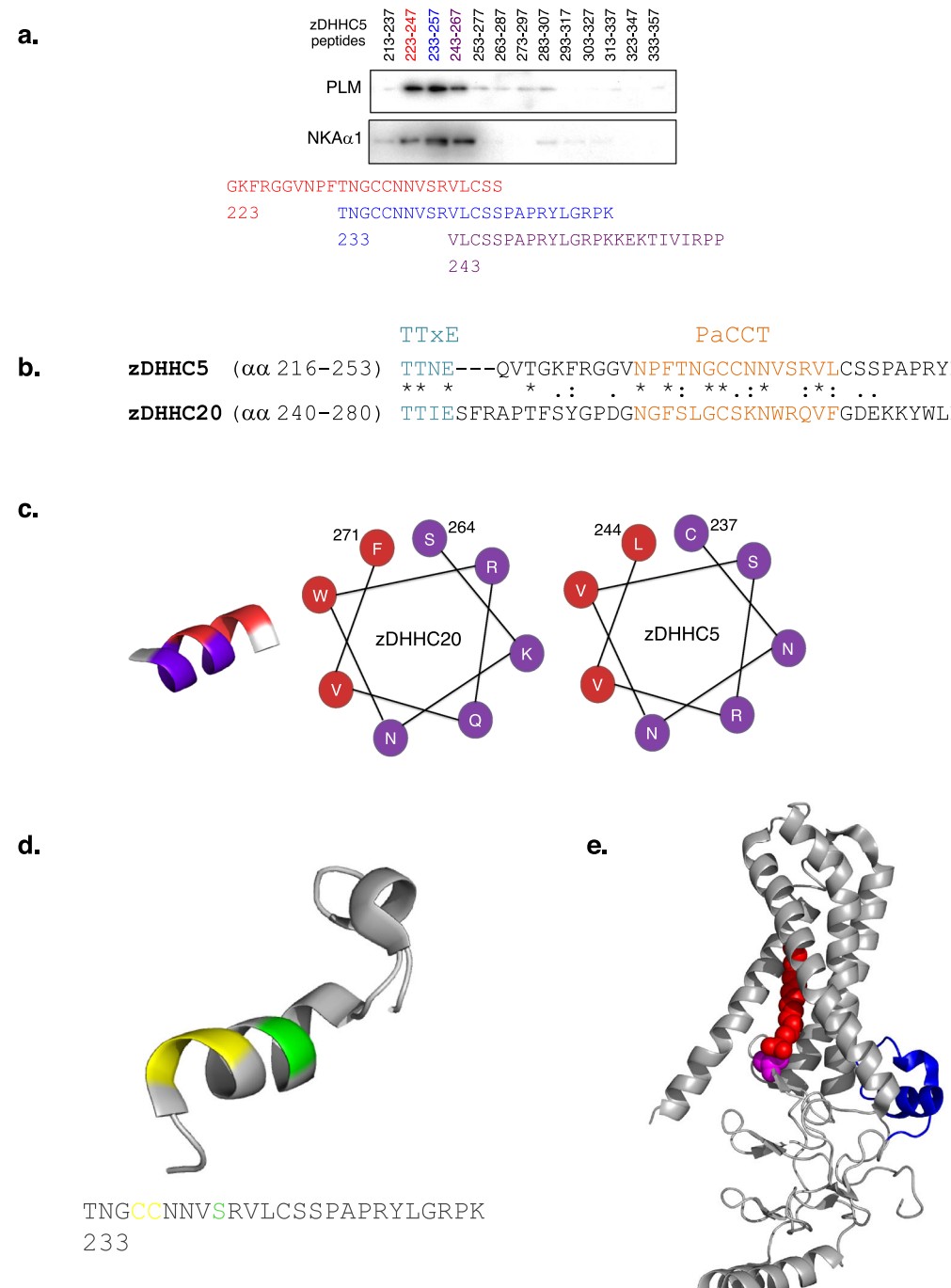

**Fig. 1 Identification of the Na-pump binding site on zDHHC5. a** The Na-pump was purified from rat ventricular lysate using an array of overlapping peptides covering zDHHC5 region αα 218–334. A binding peak comprising three overlapping peptides centered on peptide αα 233–257, the Na-pump binding site peptide (NBSP), was observed. **b** Alignment of the zDHHC5 and 20 C-terminal domains in the region of their TTxE and PaACT motifs. **c** The PaACT motifs of both zDHHC5 and 20 contain an amphipathic helix (red, non-polar; purple, polar). **d** Structural model of the NBSP based on the structure of zDHHC20 (PDB accession code 6BML) with putative sites of post-translational modification shown (palmitoylation, yellow; GlcNAcylation, green). **e** The position of the NBSP (blue) on the structure of zDHHC20 (PDB 6BML). The NBSP is located a considerable distance from the enzyme active site (C of zDHHC motif, pink; 2-bromopalmitate, red) suggesting that the interaction between zDHHC5 and PLM is likely to be indirect. Figure 1d, e were made using MacPYMOL.

An antibody specific to PLM phosphorylated at serine 68 (which captures PLM as part of the Na-pump complex[38]) was added to the other half of each lysate, and interacting proteins purified using protein G sepharose beads. The Na-pump α subunit could only be co-immunoprecipitated with (phospho-S68)PLM from the C12E10 lysate suggesting that Triton, SDS, and FC-12 had caused dissociation of the Na-pump oligomer into individual subunits, potentially in a partial or fully denatured state (Fig. 2b). The NBSP purified the pump α subunit but not PLM from the FC-12 lysate suggesting that zDHHC5 interaction with the Na-pump is independent of the presence of PLM, and is likely through a direct interaction with the α subunit (Fig. 2b).

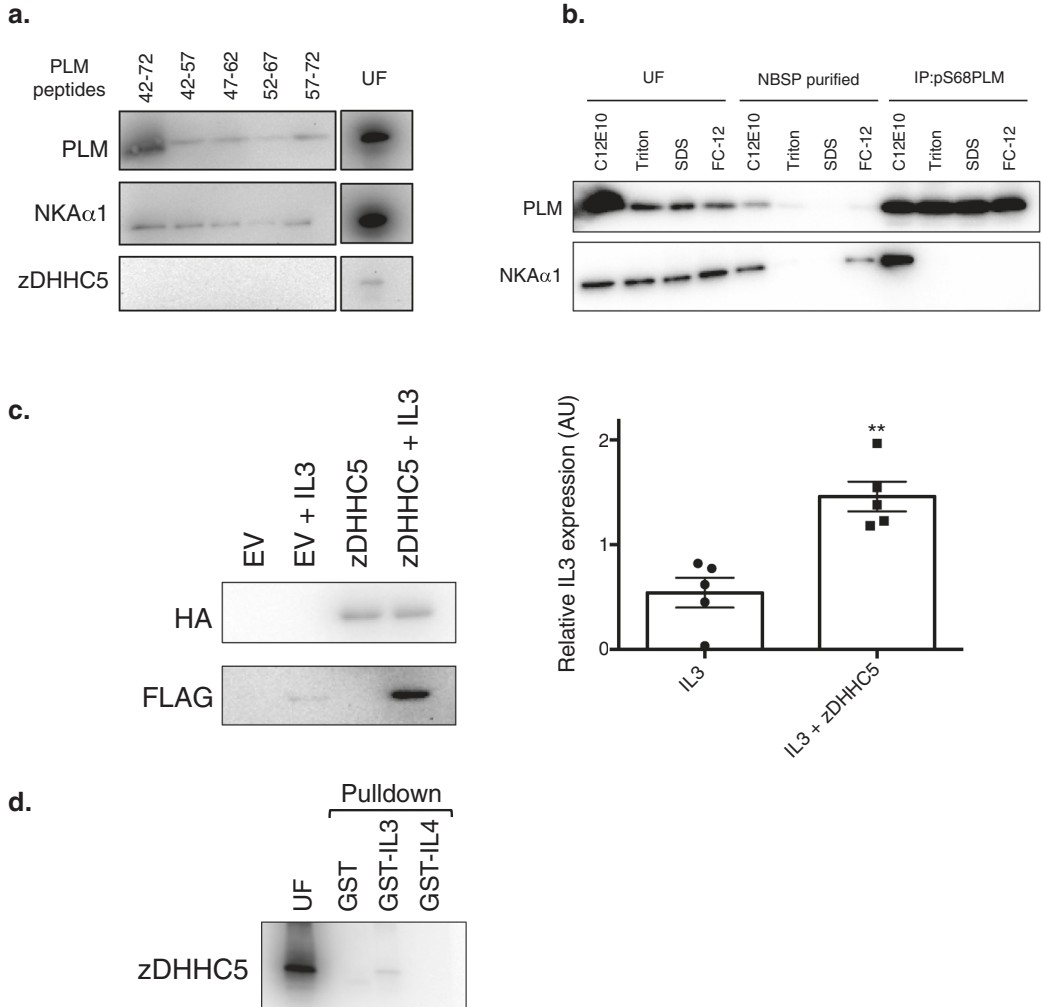

**Fig. 2 zDHHC5 interacts with intracellular loop 3 of the Na-pump α-sub-unit. a** A series of peptides covering the PLM C-tail successfully purified endogenous Na-pump α sub-unit but not zDHHC5 from rat ventricular myocyte lysate. **b** Whole rat heart homogenate was lysed with four different detergents (C12E10, Triton X-100, SDS, and FC-12). Each lysate was split in half and two separate purifications performed using the NBSP or an antibody specific to PLM phosphorylated at serine 68. **c** Co-expression of HA-zDHHC5 in FT-293 cells stably expressing human PLM caused an increase in FLAG-IL3 accumulation compared to when FLAG-IL3 was expressed alone ($n = 5$: ** $p < 0.01$). **d** HA-tagged zDHHC5 can be purified from a HEK cell lysate using recombinant GST-IL3 but not GST alone or GST-IL4.

We sought to identify the region of the Na-pump α sub-unit that interacts with zDHHC5. Co-expression of zDHHC5 with the third intracellular loop of the pump α1 sub-unit (IL3) in HEK cells resulted in a significant stabilization of IL3 (human isoform: αα 339–772) (Fig. 2c) which suggests that zDHHC5 and IL3 interact. Furthermore, GST-tagged recombinant IL3 but not GST alone nor GST fused to the fourth intracellular loop of the α subunit (GST-IL4 (αα 824–843)) purified HA-tagged zDHHC5 from a HEK cell lysate (Fig. 2d). We conclude that IL3 is the region of the Na-pump α subunit that interacts with zDHHC5.

**zDHHC5 acylation by zDHHC20 regulates Na-pump recruitment.** The region of zDHHC5 that interacts with the Na-pump contains three cysteine residues located at positions 236, 237, and 245. To determine whether or not these cysteines are acylated, their palmitoylation status in HEK cells was investigated by both resin-assisted capture of acylated proteins (acyl RAC, which affinity purifies palmitoylated proteins)[39] and acyl PEG exchange (APE, which exchanges palmitate for a 5 kDa PEG molecule and hence induces a band shift on SDS-PAGE)[13,40]. As the catalytic cysteine in zDHHC-PATs is transiently palmitoylated during the reaction cycle[18], and acyl RAC purifies palmitoylated proteins regardless of whether they are singly or multiply palmitoylated, the palmitoylation of catalytically inactive zDHHS5 (C134S) was assessed. zDHHS5 is robustly palmitoylated in HEK cells (Fig. 3a). Mutation of cysteines 236 and 237 to alanine, however, largely abolished zDHHS5 palmitoylation indicating that either and/or both residues are modified. Mutation of cysteine 245 was essentially without effect, suggesting that this residue is not palmitoylated. We suggest that the residual palmitoylation observed with zDHHS5 C236A/C237A reflects palmitoylation of zDHHC5 in its cysteine-rich domain similar to that which occurs in other zDHHC enzymes[41]. Acyl PEG exchange revealed that zDHHC5, zDHHS5, C236A/C237A zDHHS5, and C245A zDHHS5 had 3, 2, 1, and 2 palmitoylated cysteines in HEK cells respectively, suggesting that either C-236 or C-237 is palmitoylated (Fig. 3b). Mutation of C236 and C237 to alanine also reduced co-immunoprecipitation between zDHHC5 and the Na-pump (Fig. 3c), suggesting that the presence of a membrane anchor in this part of zDHHC5 contributes to its interaction with the Na-pump.

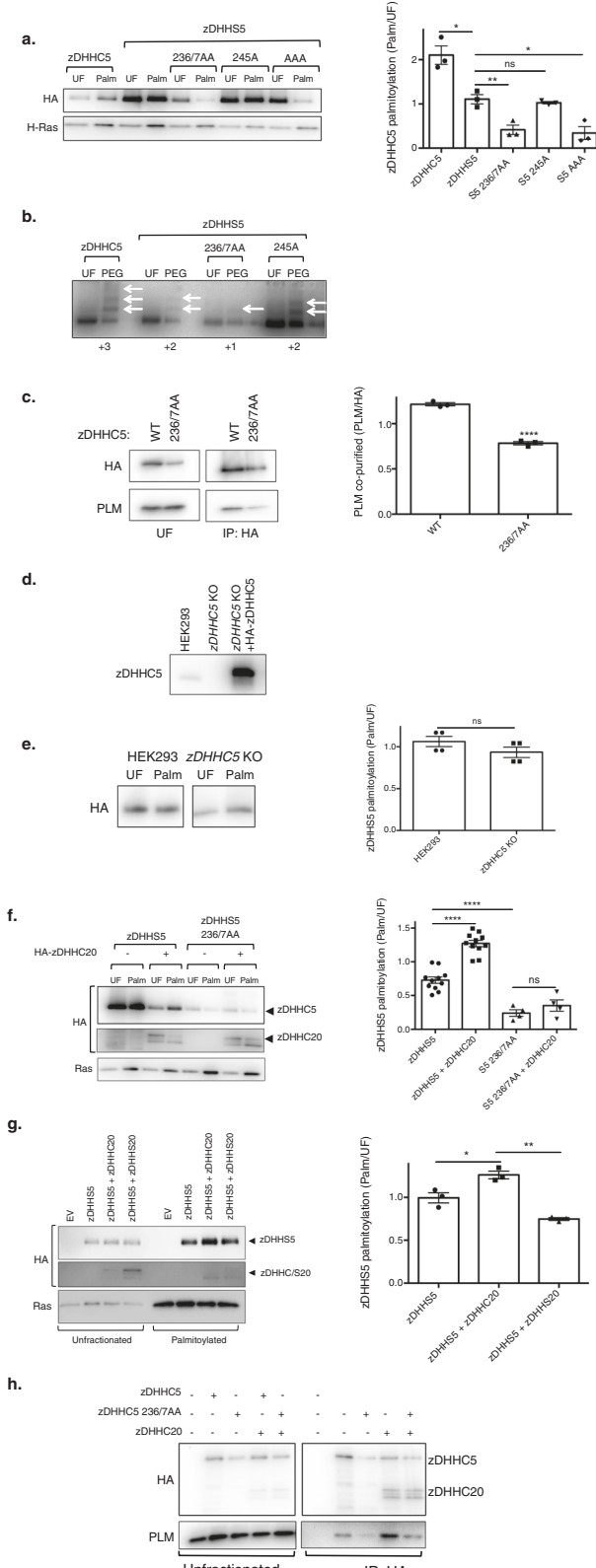

**Fig. 3 Na-pump recruitment to zDHHC5 can be regulated by palmitoylation. a** HEK 293 cells were transiently transfected to express WT DHHC5 or catalytically inactive DHHS5 +/− C236A/C237A, (AA), C245A, (A) or C236A/C237A/C245A, (AAA) mutants. Palmitoylated fractions were purified by resin-assisted capture (Acyl-RAC) and visualized by Western Blot. The average data is shown in graph form with palmitoylation expressed as the ratio of Palmitoylated (Palm)/ Unfractionated (UF) zDHHC/S5 ($n = 3$: *$p < 0.05$, **$p < 0.01$). **b** The number of sites palmitoylated in zDHHC5, zDHHS5, and three zDHHS5 C-tail mutants (236/7AA, 245A and 236/7–245AAA) was determined by Acyl PEG exchange. **c** Co-immunoprecipitation of HA-tagged zDHHC5 and 236/7AA mutant with PLM. The amount of PLM co-immunoprecipitated was expressed relative to the amount of zDHHC5 immunoprecipitated ($n = 3$: ****$p < 0.0001$). **d** HEK 293 cells were engineered with CRISPR/ Cas9 to create a zDHHC5 knockout cell line (zDHHC5 KO). zDHHC5 can be reintroduced to the KO cells by transient transfection. **e** Both HEK 293 and zDHHC5 KO cells were transiently transfected to express zDHHS5. The bar chart shows relative palmitoylation (Palm/UF) of zDHHS5 as determined by Acyl-RAC ($n = 3$). **f** zDHHS5 and C236/7AA-zDHHS5 were expressed in HEK cells +/− zDHHC20. The palmitoylation status (Palm/ UF) of each zDHHS5 mutant was determined by Acyl-RAC. ($n = 4$: *$p < 0.05$, **$p < 0.01$). **g** Impact of wild type and catalytically inactive zDHHC20 on the palmitoylation status of zDHHS5 ($n = 4$: *$p < 0.05$, **$p < 0.01$). **h** HA-tagged WT zDHHC5 and the C236/7AA mutant were expressed in HEK 293 cells +/− zDHHC20 and immunoprecipitated with HA beads. Associated PLM was visualized by Western Blotting.

(Fig. 3e). To identify the zDHHC-PAT responsible we used a proximity biotinylation approach. This employs a mutant (R188G) form of the bacterial biotin ligase BirA that releases a highly reactive, labile reaction intermediate which biotinylates primary amines on nearby (interacting) proteins[42]. We generated fusion proteins with BirA fused at either the intracellular N or C terminus of zDHHC5, and biotinylated proteins interacting with the zDHHC5-BirA fusions (or BirA alone) in HEK cells identified by LC-MS/MS following their purification with streptavidin beads (Supplementary Data 1). The only zDHHC-PAT found to interact with zDHHC5 was zDHHC20 (Supplementary Data 2). Overexpression of HA-tagged zDHHC20 (Fig. 3f) but not DHHS20 (Fig. 3g) caused increased palmitoylation of the zDHHS5 C-tail. Furthermore, zDHHC20 overexpression also caused increased co-immunoprecipitation of zDHHC5 with the Na-pump in HEK cells (Fig. 3h). Hence, zDHHC20 palmitoylates zDHHC5 at C236/7, regulating the interaction between zDHHC5 and the Na-pump, controlling PLM palmitoylation.

**zDHHC5 O-GlcNAcylation regulates PLM palmitoylation.** *O*-GlcNAcylation—the attachment of *O*-linked *N*-acetylglucosamine (*O*-GlcNAc) sugar groups to intracellular proteins—is a critically important post-translational modification that regulates a wide range of cellular processes in metazoans[43]. The reversible attachment of *O*-GlcNAc to specific serine and threonine residues in particular substrate proteins is carried out by a single pair of enzymes, called *O*-GlcNAc transferase (OGT) and *O*-GlcNAcase (OGA). Using proximity biotinylation we found that OGT interacts with both BirA-zDHHC5 and zDHHC5-BirA in the cardiomyoblast cell line H9c2 (Supplementary Data 3). In zDHHC5, S241 is located on the hydrophilic face of the amphipathic helix (Fig. 1d), and fits the OGT consensus sequence (relative to the GlcNAcylated serine position −1, V; +1, R[44]). OGT substrates are typically disordered on the C terminal side of the GlcNAcylation site[44]; the zDHHC5 C tail is predicted to be disordered[13,45]. This suggested to us that GlcNAcylation may regulate Na-pump recruitment to and PLM palmitoylation by

To determine whether the dicysteine motif is palmitoylated by endogenous zDHHC5 or another upstream zDHHC-PAT, the palmitoylation status of HA-tagged zDHHS5 in wild type and *zDHHC5* knockout (Fig. 3d) HEK cells was determined. The catalytically inactive enzyme had essentially identical palmitoylation in both cell lines, suggesting that the zDHHC5 dicysteine motif is palmitoylated by an enzyme other than zDHHC5

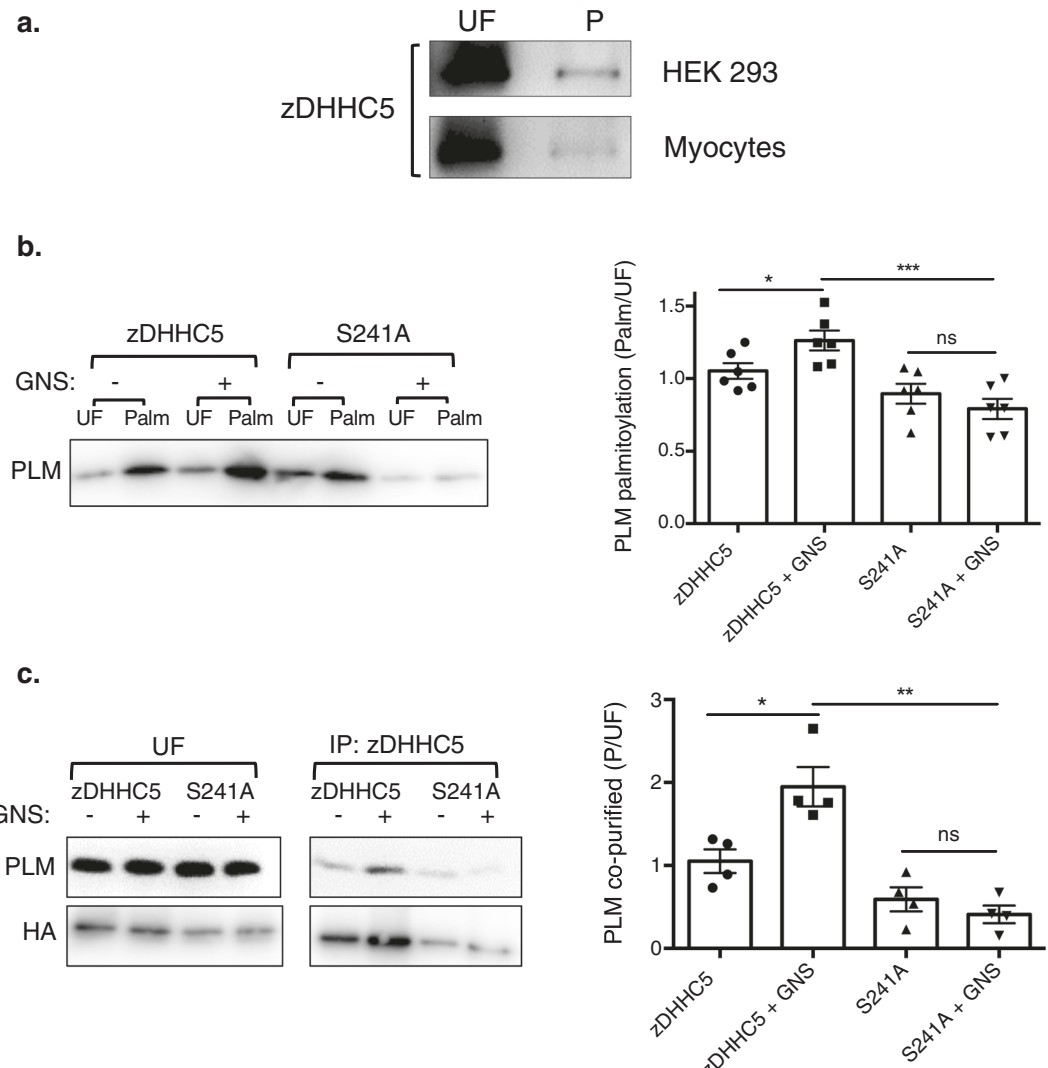

**Fig. 4 zDHHC5 GlcNAcylation regulates PLM palmitoylation. a** Affinity purification of zDHHC5 with recombinant HALO-tagged catalytically inactive OGA from both HEK cells (upper) and adult rat ventricular myocytes (lower). UF unfractionated lysate, P OGA purified fraction. **b** A FT-293 stable cell line expressing human PLM was transfected with HA-tagged zDHHC5 or a GlcNAcylation-deficient mutant (S241A) and treated overnight $+/-$ 1 μM GNSG (an OGA inhibitor). Relative PLM palmitoylation (Palm/UF) was determined by acyl-RAC ($n = 6$: $*p < 0.05$, $***p < 0.001$). **c** From the same unfractionated samples, zDHHC5 was also immunoprecipitated using HA beads, and the amount of associated PLM determined by Western Blotting ($n = 4$: $*p < 0.05$, $**p < 0.01$).

zDHHC5, and this possibility was considered in a series of experiments. zDHHC5 could be purified with recombinant Halo-tagged catalytically inactive OGA demonstrating that the acyl-transferase is GlcNAcylated in both rat hearts and transfected HEK cells (Fig. 4a). Treatment of a FT-293 stable cell line expressing PLM with the OGA inhibitor GlcNAc-statin (GNS) increased both PLM palmitoylation and co-immunoprecipitation between zDHHC5 and the Na-pump in cells transfected with wild-type zDHHC5 but not a S241A mutant (Fig. 4b, c). Together, this suggests that GlcNAcylation of zDHHC5 S241 promotes PLM palmitoylation through increased recruitment of the Na-pump to zDHHC5.

**Pharmacological manipulation of PLM palmitoylation.** Site-specific palmitoylation and GlcNAcylation of zDHHC5 promotes PLM palmitoylation through increased recruitment of the Na-pump to the acyltransferase. Next, we wanted to ascertain whether or not PLM palmitoylation could be reduced with a pharmacological tool by blocking the interaction between

zDHHC5 and the Na-pump. HEK cells stably expressing human PLM were incubated with a cell-penetrating stearate-tagged version of NBSP. Following both 3 and 24 h treatments, PLM palmitoylation was significantly reduced in cells treated with stearate-tagged NBSP compared to an equivalent scrambled control (Fig. 5a). In contrast, the disruptor peptide had no effect on H-ras palmitoylation, which was expected as H-ras is a substrate for zDHHC9[46]. The disruptor peptide also reduced the palmitoylation of endogenous PLM but not Caveolin-3 in rat ventricular myocytes (Fig. 5b). In short, our results suggest that PLM palmitoylation can be selectively reduced using a pharmacological tool that prevents the Na-pump from interacting with zDHHC5.

## Discussion

Although protein S-palmitoylation is a critical regulator of cellular function, the molecular mechanism by which substrates are recognized and enzyme catalysis occurs is only poorly understood. Here, we have identified a discrete binding site on zDHHC5 for the Na-pump. We find that post-translational

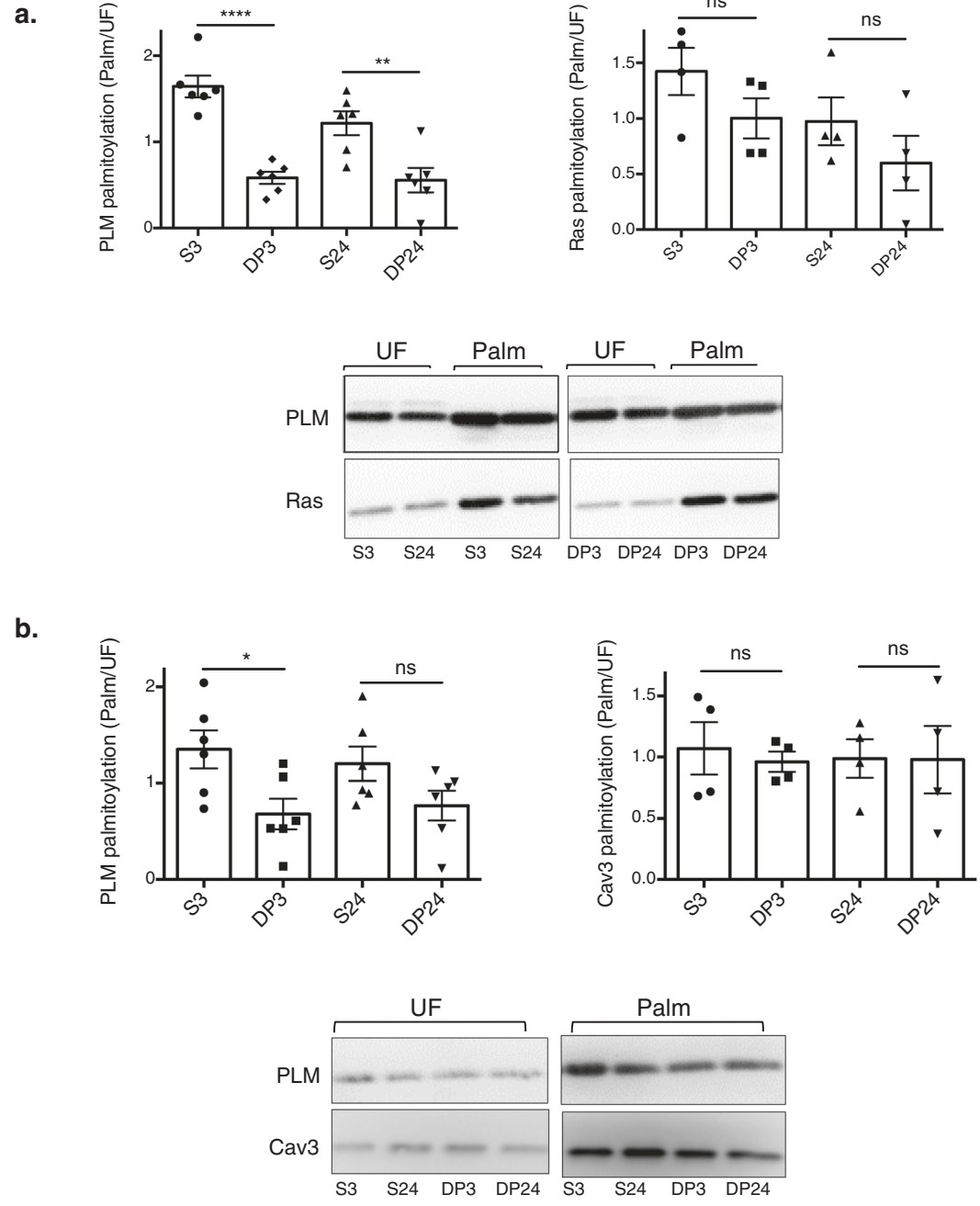

**Fig. 5 Selective pharmacological manipulation of PLM palmitoylation. a** HEK cells stably expressing human PLM were incubated with 30 μM of a stearate-tagged peptide (either NBSP (DP) or scrambled NBSP (S)) for a period of 3 or 24 h. Following cell lysis, relative PLM and Ras palmitoylation (Palm/UF) values were determined using a combination of Acyl-RAC and Western Blotting (PLM: $n = 6$; Ras: $n = 4$; **$p < 0.01$, ****$p < 0.0001$). **b** The same treatments were also applied to adult rat ventricular myocytes prior to determining the palmitoylation status of both PLM and Cav3 (PLM: $n = 6$; Cav3: $n = 4$; *$p < 0.05$).

modifications near this binding site regulate substrate recruitment, and that disrupting the zDHHC5-Na-pump interaction reduces palmitoylation of the zDHHC5 substrate PLM. Hence we demonstrate the selective pharmacological targeting of a palmitoylated protein by interfering with its recruitment to a zDHHC-PAT.

The Na-pump binding site on zDHHC5 is located distal to the enzyme active site and contacts the third intracellular loop (IL3, αα 339–772) of the Na-pump α subunit. This interaction correctly positions PLM within the zDHHC5 active site, where it becomes palmitoylated. However, this does not rule out the possibility of zDHHC5 and the Na-pump α-subunit interacting indirectly via

an adapter protein. Recently the activity and localization of zDHHC5 in HEK and HeLa cells has been found to be regulated by two closely related proteins, Golga7[47] and Golga7b[23] respectively. Golga7b is palmitoylated by zDHHC5 and interacts with the same region of the zDHHC5 C tail as the Na-pump in a palmitoylation-dependent manner. Palmitoylated Golga7b is required for delivery of zDHHC5 to the cell surface as well as recruitment to and palmitoylation of substrates by zDHHC5[23]. When we carried out a detailed proteomic analysis of the proteins in cardiac muscle that interact with NBSP, however, neither Golga7 nor Golga7b were identified[48]. Therefore, if an adapter protein is required to mediate the interaction between zDHHC5

and the Na-pump in the heart, it is unlikely to be either Golga7 or 7b but rather an as yet unidentified protein of novel function.

A major finding of our work is the observation that covalent modification (both palmitoylation and GlcNAcylation) of residues proximal to a substrate-binding site on a zDHHC-PAT affected both the recruitment and palmitoylation of that substrate—a phenomenon that will likely apply to many other zDHHC-PATs and enzyme-substrate combinations. Regulation of substrate recruitment by palmitoylation of cysteine residues in the zDHHC5 C-tail has previously been postulated by Yang and co-workers[37], and is now confirmed here. The C-terminal regions of all zDHHC-PATs contain one or more cysteines suggesting that substrate recruitment to zDHHC-PATs may commonly be regulated by site-specific palmitoylation. Our studies also revealed the existence of a palmitoylation cascade with zDHHC20 acting upstream of zDHHC5. The ER-resident zDHHC-PAT zDHHC6 is palmitoylated by zDHHC16 at three cysteine residues located within a SH3 binding domain at the enzyme's C-terminus[49]. It would, therefore, seem likely that palmitoylation cascades are a common regulatory feature of zDHHC-PAT activity also.

Palmitoylation of the cardiac sodium/calcium exchanger (NCX1) is determined by the presence of a juxtamembrane amphipathic helix located next to the cysteine residue that gets modified[50]. By analogy, it is possible that palmitoylation at the dicysteine motif on zDHHC5 by zDHHC20 is facilitated by the zDHHC5 amphipathic helix, with specificity for zDHHC20 determined by the chemical properties of the side chains protruding from the hydrophilic face of the helix (as is the case for NCX1). Palmitoylation of the ubiquitin ligase GobX by zDHHC20 is also dependent on an amphipathic helix proximal to the palmitoylated cysteine[51]. In contrast, the zDHHC20 palmitoylation sites in the epidermal growth factor receptor lie in an unstructured part of its C tail[52].

In contrast to the relationship between some zDHHC-PATs and their substrates[53], the interaction between zDHHC5 and the Na-pump was sufficiently tight to enable their co-purification in complex with each other. One question arising from this observation is by what means is the Na-pump released from zDHHC5 following palmitoylation of PLM? We anticipate that the substrate release is driven by the energetic requirement of the fatty acid attached to PLM to reside within a hydrophobic membrane rather than aqueous solution. We speculate that incorporation of the fatty acid within the lipid bilayer will induce conformational changes within the Na-pump, PLM in particular, that result in release of the pump from zDHHC5.

Specific manipulation of protein palmitoylation with small molecules for use in the clinic is highly desirable but currently remains out of reach[54]. To date, efforts to discover inhibitors of zDHHC-PATs have exclusively focussed on the enzyme's active site as the catalytic pocket is considered to be druggable. Even if isoform-specific active site inhibitors could be identified, however, they are unlikely to be therapeutically useful as they would block palmitoylation of the entire substrate ensemble of the zDHHC-PAT(s) on which they act, not just the protein of interest. The data presented here, however, suggests that in future it may be possible to develop molecular modulators of protein palmitoylation that act by selectively blocking or enhancing the recruitment of particular substrate proteins to their cognate zDHHC-PAT(s). For example, a molecule that prevented recruitment of the Na-pump to zDHHC5 would block PLM palmitoylation increasing the pump's activity, and would be useful for treating heart failure and cardiac hypertrophy[31]. Molecules that target substrate binding sites on zDHHC-PATs themselves may suffer from a lack of selectivity if several substrates bind to the enzyme in the same region. For example, the

disruptor peptide used here may also alter the palmitoylation status of other proteins that interact with the NBSP region of zDHHC5. In contrast, molecules that interact with the substrate itself blocking or enhancing its interaction with (a) zDHHC-PAT (s) offer better promise. Protein S-palmitoylation is associated with a wide-range of cancers[52,55–60], and can be hijacked by pathogens[61,62]. Through understanding how individual proteins are recruited to their partner zDHHC-PAT(s), it should be possible to develop screening strategies that will enable the identification of molecules that selectively alter the palmitoylation status of particular proteins that can ultimately be developed into therapeutic drugs for use in the clinic to treat a wide-range of diseases.

## Methods

**Ethics statement**. This study utilized primary ventricular myocytes from male Wistar rats (250–300 g, 7–9 weeks). All protocols involving animals were approved by the Animal Welfare and Ethics Review Boards at the Universities of Dundee and Glasgow. Rat cardiac tissues were collected postmortem after sacrificing animals using a method designated Schedule 1 by the Animals (Scientific Procedures) Act 1986.

**Antibodies and chemicals**. All chemicals were of the highest grade available and were purchased from Sigma–Aldrich unless indicated otherwise. All HEK cell transfections utilized Lipofectamine 2000 (ThermoFisher Scientific). Primary antibodies were from the following sources: PLM phospho-S68—custom made[38], zDHHC5—Sigma–Aldrich HPA014670 (diluted 1:2000 for western blots), Flotillin 2—BD Biosciences 610383 clone 29 (1:2000), PLM—Abcam ab76597 (1:2000), HA tag—Roche, clone 3F10 (1:2000), FLAG—Sigma F1804 clone M2 (1:2000), Caveolin 3—BD Biosciences 610421 clone 26 (1:5000), sodium pump α1 subunit, Development Studies Hybridoma Bank, clone α6F (1:100), Ras—Merck 05-516, clone RAS10 (1:2000). HRP-linked secondary antibodies were from ThermoFisher Scientific (antirat, diluted 1:2000 for western blots) and Jackson ImmunoResearch (antirabbit and antimouse, both diluted 1:2000 for western blots). Western Blots utilized Immobilon Western Chemiluminescent HRP Substrate (Merck Millipore, Watford, UK) and were visualized and analyzed using a ChemiDoc XRS acquisition system running QuantityOne software (BioRad).

**zDHHC5 peptide array**. An array of 25 amino acid peptides overlapping by 15 residues each, representing the C tail of rat zDHHC5 was purchased from Alta Bioscience. Each peptide had an N terminal biotin for affinity purification. Rat ventricular lysates were precleared by incubation with streptavidin Sepharose (GE Life Sciences) for 1–2 h at 4 °C, incubated with 1 µg peptide (~0.2 µM final concentration) overnight at 4 °C with end-over-end mixing, and the peptides purified the next day with streptavidin Sepharose. After extensive washing peptide–protein complexes were eluted with SDS PAGE loading buffer.

**Plasmids and transfections**. Plasmids encoding HA-tagged murine zDHHC isoforms were generously provided by Masaki Fukata, Division of Membrane Physiology, National Institute for Physiological Sciences, Okazaki, Aichi, Japan.
   HEK-derived FT-293 cells constitutively expressing human PLM were generated using the Invitrogen Flip-In system. All transfections of plasmid DNA used Lipofectamine 2000 (Invitrogen) according to the manufacturer's instructions.

**Site-directed mutagenesis**. All point mutants were generated using the Quik-Change II site directed mutagenesis kit (Agilent). Mutagenesis primers were designed using the QuickChange primer design online tool provided by Agilent, and purchased from ThermoFisher Scientific. Mutagenesis primers were zDHHC5 C236/7AA: forward—tgaatcccttcaccaatggcgccgctaacaacgttagccgtgtcc, reverse—ggacacggctaacgttgttagcggcgccattggtgaaggggattca; zDHHC5 S241A: forward—ccaatggctgctgtaacaacgttgcccgtgtcctctg, reverse—cagaggacacgggcaacgttgtta-cagcagccattgg; zDHHC5 C245A forward—gttagccgtgtcctcgccagttctccagcacc, reverse—ggtgctggagaactggcgaggacacggctaac; zDHHC20 C156S (zDHHS20) forward—tcttaa-gatggatcatcacagtccatgggtgaataactg, reverse—cagttattcacccatggactgtgatgatccatcttaaga.

**Immunoprecipitation and pulldown experiments**. We used anti HA affinity matrix (Roche) to immunoprecipitate HA tagged proteins from cell lysates as described previously[13].
   GST fusion proteins of the Na-pump third and fourth intracellular loops (human isoform: IL3 (aa 339–772), IL4 (aa 824–843) were expressed in *E. coli* strain BL21(DE3) and purified using a 5 ml glutathione column (GE Healthcare). Following desalting into PBS, the recombinant proteins were incubated overnight with HEK cell lysates that had been prepared in PBS supplemented with 2 mg/ml

C12E10 and protease inhibitors. The following day, interacting proteins were captured by pulling down with glutathione sepharose beads (GE Healthcare).

Halo-tagged catalytically inactive (D298N) O-GlcNAcase (OGA) from *Clostridium perfringens* was used to purify GlycNAcylated proteins from HEK cell and ventricular myocyte lysates via HaloLink agarose (Promega) in RIPA buffer (50 mM Tris pH 7.5, 1% NP-40, 0.5% sodium deoxycholate, 0.1% SDS, 150 mM NaCl, 2 mM EDTA, 50 mM sodium fluoride)[63]. After extensive washing (50 mM Tris pH 7.5, 0.02% TWEEN 20, 450 mM NaCl) captured proteins were eluted by incubating the beads for 45 min at 4 °C with wash buffer supplemented with 2 mM Thiamet G.

**Adult rat ventricular myocytes.** Calcium-tolerant adult rat ventricular myocytes were isolated by retrograde perfusion of collagenase in the Langendorff mode. Myocytes were plated onto laminin coated culture dishes before experimentation.

**Resin assisted capture of acylated proteins.** Acylated proteins were purified using thiopropyl Sepharose (GE Life Sciences) in the presence of neutral hydroxylamine from cell lysates in which irrelevant cysteines had first been alkylated with S-Methyl methanethiosulfonate[38,39]. The palmitoylation status of all our proteins of interest were determined by measuring their abundance in the acyl-RAC purified fraction relative to that in the starting (unfractionated) cell lysate.

**Acyl PEG exchange.** We used a refinement[40] of our acyl PEG switch assay[13] that PEGylates previously palmitoylated cysteines following removal of hydroxylamine. Irrelevant cysteines in cell lysates were blocked using 100 mM maleimide. Excess unreacted maleimide was removed by acetone precipitation, and previously palmitoylated cysteines were revealed by treatment with 250 mM neutral hydroxylamine for 1 h at room temperature. Lysates were desalted using Zeba spin columns (ThermoFisher Scientific) and incubated with 2 mM 5 kDa methoxypolyethylene glycol maleimide for 1 h at room temperature. The reaction was quenched with SDS PAGE loading buffer supplemented with β-mercaptoethanol, and analyzed directly by SDS PAGE.

**zDHHC5 knockout cells.** We used guide RNAs (gRNA) targeted against *zDHHC5* provided by Horizon Discovery to generate *zDHHC5* knockout HEK cells[64].

**Identification of biotinylated zDHHC5 interacting proteins.** HEK cells in 10 cm dishes were transfected with pcDNA3.1 mycBioID, pcDNA3.1 BirA-HA, pcDNA3.1 BirA-zDHHC5, and pcDNA3.1 zDHHC5-BirA and maintained overnight with 50 μM biotin. The next day after extensive washing cells were lysed in 0.1% SDS, 1% Triton X-100 in PBS supplemented with protease inhibitors, insoluble material removed by centrifugation, and biotinylated proteins captured using streptavidin Sepharose overnight at 4 °C. Beads were washed five times with 0.1% SDS, 1% Triton X-100 in PBS, then five times with 1 M urea in 100 mM ammonium bicarbonate. Biotinylated proteins were digested with Trypsin Gold (Promega) in 1 M urea in 100 mM ammonium overnight at 37 °C, then reductively alkylated and analyzed using a LTQ Orbitrap Velos Pro (Thermo) at the FingerPrints Proteomics Service, University of Dundee. Mascot reports including peptides identified are provided in Supplementary Data 1.

**Stearate-tagged disruptor peptides.** Peptides with a N terminal stearate group and a C-terminal Biotin were manufactured by DC Bioscience. Disruptor peptide: stearate-TNGGCCNNVSRVLCSSPAPRYLGRPK-Biotin; scrambled: stearate-LSAYNTVNRRCSKGRVPNCGPCSPLK-Biotin.

**Statistics and reproducibility.** Quantitative data are presented as means ± S.E. Differences between experimental groups were analyzed by one-way analysis of variance followed by post-hoc tests using Graphpad Prism. Differences were considered statistically significant when $p$ was <0.05.

No data was excluded from any analysis and all experiments were reproducible. In all cases sample sizes refer to biological replicates, which were defined as independently prepared, biologically distinct samples.

**Reporting summary.** Further information on research design is available in the Nature Research Reporting Summary linked to this article.

## Data availability
Reagents and datasets generated and analysed during the current study are available from the corresponding authors on reasonable request. Uncropped Western blots and source data sets are available in the Supplementary Information and Supplementary Data 4, respectively.

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

## Acknowledgements

We thank Dr. Christophe Lachaud for providing advice and materials to generate the zDHHC5 knockout cell line and Professor Daan van Aalten for reagents to measure and manipulate O-GlcNAcylation. This work was supported by grants from the British Heart Foundation (FS/14/68/30988 to W.F. and N.F., RG/17/15/33106 to W.F., N.F. and M.J.S.).

## Author contributions

F.P., J.H., J.K., E.B., and N.J.F. conducted experiments and analyzed the data. W.F., N.J.F., and M.J.S. conceived the idea for the project. W.F., N.J.F., and J.H. wrote the paper.

## Competing interests

The authors declare no competing interests.
