## [Peer Review File · Communications Biology]

Reviewers' comments:

Reviewer #1 (Remarks to the Author):

Summary: The major findings of the manuscript are that the juxtamembrane amphipathic helix of zDHHC5 associates with the alpha subunit of the sodium pump to facilitate palmitoylation of its associated regulatory subunit phospholamman (PLM). Modification of zDHHC5 with palmitate at cysteine residues adjacent to the amphipathic helix is mediated by zDHHC20. Similarly, this same region of zDHHC5 is modified with O-GlcNAc by OGT. Palmitoylation and O-GlcNAcylation both increase association of the Na pump with the enzyme that is accompanied by an increase in palmitoylation of PLM. Delivery of a TAT-tagged peptide that mimics the zDHHC5 binding site into cells reduces PLM palmitoylation, suggesting the potential to interfere with palmitoylation of specific proteins by disrupting the enzyme's protein:protein interactions with its substrates.

Overall impression of the work: There is considerable interest in and justification for identifying strategies to interfere with palmitoylation of specific proteins as a potential therapeutic intervention. This manuscript is the first to address this issue by looking at zDHHC5 and its substrate PLM. The manuscript builds on prior work (ref. 13) that clearly demonstrated that PLM is a zDHHC5 substrate and the importance of the juxtamembrane region in the C-terminal domain of zDHHC5 for substrate recruitment. That paper implicated direct binding of PLM to the juxtamembrane region. Guided by the first structure of a zDHHC protein, the authors now show that the Na pump alpha subunit is binding to zDHHC5 in this region, which can be modeled as an amphipathic alpha helix. The association of the alpha subunit with zDHHC5 then presumably positions PLM closer to the enzyme active site where it is palmitoylated. Strengths of the study include the mass spec analyses that identified the association of zDHHC20 and OGT with zDHHC5, enabling characterization of zDHHC5 PTMs that modify Na pump association and PLM palmitoylation. Enthusiasm for the manuscript is diminished by the absence of controls for the TAT-peptide experiment, a lack of clarity and missing information in the text, and problems with data presentation as outlined below.

Specific Comments

1. The "Na-pump" needs to be clearly described in the introduction, particularly if the manuscript is to be accessible to a broad readership. Does this refer to an oligomeric complex of alpha and beta subunits and PLM?
2. The authors acknowledge in the discussion that they cannot exclude the possibility of another protein mediating binding of the Na pump alpha subunit to zDHHC5. Is the Na pump beta subunit a possible candidate? This could be addressed experimentally or a clear justification why it should not be considered should be presented in the text.
3. Figure 2b: What is the rationale for immunoprecipitating with a phospho-S68 PLM antibody vs. an antibody for PLM? Does this capture a unique population of the Na pump?
4. Figure 2b: The source and characterization of the phospho-S68 PLM antibody is not provided in the text.
5. Figure 2b: The text describing this figure could be presented more clearly. The NBSP pulldown of the presumably intact complex with C12E10 should be mentioned, lending more significance to the finding that FC-12 disrupts the complex.
6. Many of the conclusions of the manuscript rely on quantitation of the western blots. Information

was not provided on the method of detection (fluorescence or chemiluminescence?) and quantitation. Importantly, what was done to assess whether the quantitation was performed in a linear range of detection? The y-axis of the bar graphs do not have units.

7. The number of independent experiments that were quantified were not shown. Does the standard error in the bar graph represent multiple independent experiments or are these replicate lanes?

8. Figure 3c: The unfractionated HA blot shows less zDHHC5 236/237AA than WT in the lysate. Can this account for the reduced amount of PLM seen in the co-IP vs. a reduction in the interaction?

9. The specificity of co-IPs of integral membrane proteins is a concern. To argue that Figure 3f confirms is the only zDHHC protein to interact with zDHHC5, the authors should overexpress another DHHC protein and show that it doesn't pulldown zDHHC5. zDHHC2 is a reasonable candidate as it shows some plasma membrane staining when overexpressed. Alternatively, Figure 3f could be removed from the manuscript.

10. Figure 4a is not very convincing. The signal in the P boxes is weak and confidence is not increased when the gel is cropped to a single band.

11. Figure 4c – the bands are continuous, raising questions about the quality of the data used for quantitation.

12. The TAT peptide experiment needs an additional control to show that treatment of the cells with a TAT peptide is not impacting PLM palmitoylation. One suggestion is to use a scrambled NBSP sequence, although there might be better controls described in the TAT peptide literature. Quantitation of the Ras palmitoylation should also be presented. The same issues described earlier (items 6 and 7) are particularly important here as this experiment is the punchline of the paper.

13. Experimental Procedures: zDHHC5 peptide array – what are the conditions for solubilization of the rat ventricular lysates? What detergent was used and how was insoluble material removed? Incomplete solubilization of membrane proteins can result in small membrane fragments that are only pelleted at high g force.

Reviewer #2 (Remarks to the Author):

In this manuscript, Plain et al. identify an amphipathic helix that mediates interaction between zDHHC5 and the α - subunit of the Na pump. The interaction is tight enough that the complex between the peptide and the Na pump can be purified. The authors identify posttranslational modifications adjacent to the amphipathic helix of zDHHC5 to be important determinants in modulating the interaction between zDHHC5 and the Na pump. Finally, they show that by using a peptide with the sequence of the binding region on zDHHC5, they were able to perturb the interaction between zDHHC5 and the Na pump, which was reflected in decrease in PLM palmitoylation. Although zDHHC enzymes have been the subject of an increasing number of studies, identification of a discrete binding interface between a zDHHC enzyme and the cognate substrate is a key missing piece from the literature in this field. Moreover, chemical probes to manipulate zDHHC activity is sorely missing and the ability to do so, albeit with a peptide, constitutes an important starting point, particularly since an enzyme-substrate interaction was successfully targeted and not generically the active site of the enzyme.

I recommend publication in Communications Biology.

A couple of experiments will strengthen the paper -

1) Can the authors use crosslinking or a mutagenesis based approach to nail down the residues that are important in the enzyme-substrate interaction, i.e. between the amphipathic helix and the Na pump ?

2) In the experiments with zDHHC20 mediated palmitoylation of zDHHC5, a control experiment with a catalytically inactive mutant, such as zDHHS20 would better confirm that the catalytic apparatus of zDHHC20 is indeed responsible for palmitoylating zDHHC5.

Reviewer #3 (Remarks to the Author):

#COMMSBIO-19-0327-T - The manuscript by Fiona Plain et al. examines the regulation of zDHHC5, a DHHC palmitate transferase, a family of enzyme responsible of S-acylation of membrane proteins. The same group previously demonstrate that zDHHC5 is responsible of palmitoylation of phospholemman (PLM), an accessory subunit of the Na⁺,K⁺-ATPase (dubbed Na-pump, references 7 and 13 in this Ms). Here, they define the region of zDHHC5 that interacts with the Na-pump (NBSP). They also demonstrate that post-translational modification of zDHHC5 via palmitoylation and GlcAcylation as an impact on its interaction with the Na-pump, and therefore on PLM acylation. Finally, they show that an NBSP mimicking peptide can be used in cellulo to inhibit zDHHC5/Na-pump interaction, and thus limiting PLM palmitoylation. This Ms gives new inputs on the regulatory mechanism of DHHC palmitate transferase which remains poorly documented, while S-palmitoylation of membrane proteins is associated with a wide-range of diseases.

I have several comments for manuscript improvement. As the Ms do not contain numbering of lines, pieces of the text were cited in italic:

#1. In Figure 1A, the peptides used to fish the Na-pump are in fact pretty short, suggesting that the interaction of full-length zDHHC5 with the Na-pump is not depending on a complicate three dimensional fold. From this analysis, author pointed a particular alpha helix from the zDHHC5 enzyme covered by the sequence of the three positive peptides (Figure 1C). Additionally, the author indicate in the Discussion section that "In contrast to the relationship between some zDHHC-PATs and their substrates, the interaction between enzyme and substrate was sufficiently tight to enable purification of the Na-pump in complex with zDHHC5". Therefore, could it be considered a reciprocal experiment with peptides obtained from the Na-pump that should be used to fish zDHHC5? I strongly suggest the author to consider such an experiment. Mapping the interaction patch of zDHHC5 on the Na-pump may indeed help greatly in the understanding of the mechanism of binding to and releasing from the Na-pump. From the available structure of the Na,K-ATPase/B-subunit/PLM-like protein (as shown in fact in the figure 7 of Tulloch et al. (2011) and in the figure 4C of Howie et al. (2014)), and considering the figure 1E of the present Ms, a zone on the Na-pump should be easily defined to help in the designing and choosing of the Na-pump peptides to be tested.

#2. The interaction between the Na-pump complex and zDHHC5 seems to be strong. The structure of the Na,K-ATPase/B-subunit/PLM-like protein complex have been resolved in different conformations since the ten past years. The structure of zDHHC20 have been resolved last year, do the author consider the opportunity to decipher the structure of a Na,K-ATPase/B-subunit/PLM-like/zDHHC5 supercomplex? This proposal is only a recommendation and not a necessity for the present Ms. If zDHHC5 and zDHHC20 share a high homology, purification of decent amount of zDHHC5 can be maybe considered.

#3. A large set of techniques have been used to study protein-protein interaction in this paper. A very close parent of the Na-Pump (P20020, PMCA1) and close homologues of its accessory proteins (O75787, P54709) have been detected (Supplementary tables 1 and 2). Surprisingly, authors do not mention this result in the Discussion section. Does palmitoylation have been reported for these proteins to date?

#4. A paper in 2014 (Montigny et al., doi: 10.1074/jbc.M114.590307) and a review in 2016 (Barbot et al., doi: 10.1007/978-3-319-24780-9_10) reported the palmitoylation of Sarcolipin (SLN), an accessory regulatory peptide of the sarco-endoplasmic reticulum Ca²⁺-ATPase (SERCA). SERCA is a very close parent of the Na-Pump. Therefore, these two references may be mentioned in the introduction.

#5. Considering structural data, SLN binds to SERCA in a groove very close to the binding site of PLM on the Na-Pump. Surprisingly, PLM Cys40 and 42 are located at the membrane interface whereas SLN Cys9 is located deeply in the membrane (Winther et al. (2015), Toyoshima et al. (2015) and Barbot et al. (2016)). Do you think that the three pumps and their subunits (i.e. SERCA/SLN, Na-Pump/PLM and PMCA1) may share the same mechanism of palmitoylation?

#6. As mentioned in comment #3, a large set of very recent techniques have been used in this study to look at protein to protein interactions which is a very good point. However, as these techniques are pretty recent and for now not really common, I suggest to clearly add a few words about the principle of each of them (in the Materials and Methods section or when describing the figure in the Result section) to help readers.

#7. Several figures legends contain descriptions or conclusions that are paraphrasing the text without giving key informations on the way the experiments have been performed. These informations are also not available from the Methods section. Please avoid adding conclusions to figure legends. Replace them by a detailed description of the experiment (e.g. sample content, amount of protein loaded (if the same for each lane, it should be indicated for comparison of the level of expression in the UF fractions), detection system used, nature of the tags, nature of the antibody for co-immunoprecipitation, antibody used for detection). These parameters are absolutely necessary for the understanding of the experiment by the reader, to let them make their own analysis and conclusions. Conclusions of the authors should take place only in the result section.

#8. "Mutation of cysteine 245 was essentially without effect, suggesting that this residue is not quantitatively palmitoylated." may be replaced by "Mutation of cysteine 245 was without effect, suggesting that this residue is not palmitoylated"

#9. "suggesting that either C-236 or -237 is palmitoylated (Fig. 3b)." may be replaced by "suggesting that C-236 and/or -237 are/is palmitoylated" after deduction from the effect of the C245A mutation. Why this mutation seems to trigger multiple palmitoylation of C236 and 237?

#10. "suggesting that the presence of (a) membrane anchor(s) in this part of zDHHC5 promotes interaction with the Na-pump." The difference between WT and mutant is not strong. Maybe "promotes" could be replaced by "contributes moderately to" to make the statement more elusive. Did authors check the level of expression of the WT and mutated zDHHC5. If the level of expression of the mutated version is significantly lower than the WT, the amount of Na-pump detected will be also lower, isn't it?

#11. "Furthermore, overexpression of HA-tagged zDHHC20 caused both increased zDHHS5..." I

disagree as the level of expression decrease dramatically as observed from the UF loads? why? The level of expression of the mutant 236/7AA is very low compared to the WT and it seems to be not affected by overexpression of zDHHC20?

"...palmitoylation (Fig. 3g) and increased co-immunoprecipitation of zDHHC5 with the Na-pump in HEK cells (Fig. 3h)." Authors initially claim that zDHHC5 is not interacting directly with PLM. Here, they would like to look at a direct interaction of zDHHC5 with the Na-Pump but they show western-blot from detection of PLM, assuming that Na-pump is also present. Why not using an antibody directed against the Na-Pump to have a direct detection?

#12. "the zDHHC5 C tail is highly disordered". I do not find any structural data in ref 13 supporting this statement. I suggest writing "is predicted to be disordered" if some prediction has been done, either the program or the method used should be mentioned.

#13. "the OGA inhibitor GlcNAc-statin (GNS) increased both PLM palmitoylation and co-immunoprecipitation between zDHHC5 and the Na-pump in cells transfected with wild-type zDHHC5 but not a S241A mutant (Fig. 4b, c). Together, this indicates that GlcNAcylation of zDHHC5 S241 promotes PLM palmitoylation through..." It is of course an indirect proof as zDHHC5 has seemingly a lot of different substrates in the cell as strongly suggested by pulldown assays done by the authors (Supplementary data 1 and 2). Inhibition of its GlcNAcylation may interfere with a lot of other processes and signalling pathways in the cell that may affect indirectly the Na-pump and its accessory subunit (e.g. their level of expression). To conclude, authors need to assess precisely the gain or loss of function of the Na-pump following overexpression of WT or S241A zDHHC5. If not possible, authors should only indicate: "this suggests that..."

#14. In the Discussion section, authors wrote "We cannot, however, rule out the possibility that the zDHHC5 NBSP contains a binding site for an adaptor protein that interacts with the Na-pump a subunit (and potentially other zDHHC5 substrates too)." It sounds unlikely as nothing resembling to a putative "adaptor" was detected by MS from the different protein-protein interaction assay used during this study. In absence of any evidence, I suggest to remove this paragraph.

#15. "A major finding of our work is the observation that covalent modification (both palmitoylation and GlcNAcylation) of residues proximal to a substrate-binding site on a zDHHC-PAT affected both the recruitment and palmitoylation of that substrate – a phenomenon that will likely apply to many other zDHHC-PATs and enzyme-substrate combinations." Authors might be more careful. Since a modest modulation of the level of expression of the zDHHC5 enzyme itself (resulting from its post-translational modification) may also have an impact on the level of palmitoylation of its target, even the interaction with this target is not affected. If authors do not have any idea of the level of expression of WT and mutated zDHHC5 in the different experiments, they cannot conclude (e.g. in Fig 3A, if the amount of protein loaded on the gel is the same in each lane, there is significantly more zDHHC5 than zDHHC5 in the UF sample).

#16. "This raises the intriguing possibility that the activity of the Na-pump may be responsive to nutrient (glucose, amino acid, fatty acid and nucleotide) availability through dynamic changes in PLM palmitoylation as a consequence of zDHHC5 GlcNAcylation." This sentence should be more developed or removed. A few references may be added if authors would like to illustrate this idea. A lot of different groups have already published and demonstrate a link between diet or starvation with the expression and activity of different Na,K-ATPase in different tissues.

#17. "For example, a molecule that prevented recruitment of the Na-pump to zDHHC5 would block PLM palmitoylation increasing the pump's activity, and would be useful for treating heart failure and cardiac hypertrophy." As there is a high sequence homology between P-type cations transporters,

authors have to consider possible side effects of such a molecule. Authors have indeed detected other P-type ATPases in their pull-down assay using zDHHC5 for phishing (see comments #3).

#18. Comments on figure contents:

Figure 1:

Panel b, please add a few residue numbers to help the reader to localized the residues cited in the text.

Panel d, a caption containing the amino-acid sequence used to generate this figure, with the same colour code, should be added for sake of clarity. Where is localized the VLCSS sequence that is common to the three positive overlapping peptides in panel a? The figure legend should indicate the program used for modelling. S241 cited below should be also indicated.

Panel e, what does this structure correspond to? Is it a model of zDHHC5 based on the structure of DHHC20? Method use to generate this figure should be indicated. A few residues should be also annotated (e.g. first and last residue from the NBSP, residues from the catalytic site).

Figure 2:

Panel a, in figure legend, authors claim "A series of peptides covering the PLM C-tail successfully purified endogenous Napump α sub-unit but not recombinant HA-tagged zDHHC5 from HEK cells indicating that zDHHC5 does not interact with PLM directly.". I do not agree this statement as the peptide use only cover a small part of the PLM sequence. As PLM and zDHHC5 are transmembrane proteins, interaction can occur in the membrane or on the exoplasmic side also. No peptides covering those two regions were tested.

Panel b, lane numbers should be added on the figure and in the text to help readers. "Peptide purified" should be replaced by NBSP for sake of clarity. What was the sequence of the NBSP used here? It should be indicated in figure legend. Authors said that no PLM was fished out with NBSP in FC-12 (lane 8 starting from the left). But some PLM is in fact hardly visible (as in Triton, lane 6). As the interaction between the Na-pump and PLM do not survive to FC-12 treatment, this PLM may be fished by the NBSP, isn't it? Another fair technical reason of the presence of this band is that some sample came from the next well. To check this latter hypothesis, authors should obviously reload the same samples on a new gel but in another order. Text in the result section should be adapted.

Figure 3:

Panel a, please add lane numbers on figures and in the text as requested above. Authors should also indicate in the legend what is exactly detected with HA and with H-Ras. A few words should be added in figure legend to explain how bars were deduced from the blots.

Panel g, add lane numbers as indicated above. Why the level of expression of zDHHS5 is dramatically decreased in the presence of HA-zDHHC20 as observed when comparing UF lanes? It seems that the level of expression is not taken in account when generating bars.

Figure 4:

Panel b and c, add lane numbers.

Figure 5:

Figure legend may indicate that the untreated sample corresponds to the DMSO negative control for sake of clarity.

Reviewer #1 (Remarks to the Author):

Specific Comments

1. The “Na-pump” needs to be clearly described in the introduction, particularly if the manuscript is to be accessible to a broad readership. Does this refer to an oligomeric complex of alpha and beta subunits and PLM?

The cardiac Na pump is indeed an oligomeric complex consisting of an alpha and a beta sub-unit as well as PLM. The Introduction has been modified to detail the enzymatic function and sub-unit composition of the Na pump (page 4).

2. The authors acknowledge in the discussion that they cannot exclude the possibility of another protein mediating binding of the Na pump alpha subunit to zDHHC5. Is the Na pump beta subunit a possible candidate? This could be addressed experimentally or a clear justification why it should not be considered should be presented in the text.

We have now experimentally shown that the third intracellular loop of the pump alpha sub-unit is involved in contacting zDHHC5 (Figure 2c, 2d) although our data does not exclude the possibility that this interaction is mediated via an as yet unidentified adaptor protein. We can now, however, rule out a role for the beta subunit in contacting zDHHC5 as it is located on the other side of the pump from IL3.

3. Figure 2b: What is the rationale for immunoprecipitating with a phospho-S68 PLM antibody vs. an antibody for PLM? Does this capture a unique population of the Na pump?

*Previously, we have shown that phospho-63 PLM exists by itself as an oligomeric complex in the membrane whereas phospho-68 PLM is always found as part of the Na Pump complex (Wypijewski, K.J. et al. (2013) J. Biol. Chem. **288**, 13808-13820) . Here, the phospho-68 PLM antibody was used to ensure that we principally immunoprecipitated the Na pump complex. The text has been modified to make this point clear (page 7).*

4. Figure 2b: The source and characterization of the phospho-S68 PLM antibody is not provided in the text.

*The phospho-S68 PLM antibody is custom made and previously described in Wypijewski, K.J. et al. (2013) J. Biol. Chem. **288**, 13808-13820. The text has been modified to include these details (page 15).*

5. Figure 2b: The text describing this figure could be presented more clearly. The NBSP pulldown of the presumably intact complex with C12E10 should be mentioned, lending more significance to the finding that FC-12 disrupts the complex.

Figure Legend 2b and the main text have both been altered to make it clear that the NBSP pulls down the intact Na Pump complex in C12E10 (page 7).

6. Many of the conclusions of the manuscript rely on quantitation of the Western Blots. Information was not provided on the method of detection (fluorescence or chemiluminescence?) and quantitation. Importantly, what was done to assess whether the quantitation was performed in a linear range of detection? The y-axis of the bar graphs do not have units.

The method of detection used in the Western Blots was chemiluminescence and quantitation was performed using a Bio-Rad ChemiDoc XRS imaging system. We have established lab protocols which ensure that all measurements are not saturated and made within the linear range of detection. All recent publications from our lab have used these protocols (Example PMIDs: 28432123, 26174834, 25422474, 23532852, 21868384). With the exception of Figure 2c, the y-values recorded on the bar graphs are always a ratio between an enriched fraction (typically palmitoylated or pulled-down) relative to a corresponding unfractionated sample (UF). Therefore, the values on the y-axis all have arbitrary units. To improve the clarity of our data presentation, we have altered the labelling on the y-axes of Figures 3a, 3c, 3e, 3f, 3g, 4b, 4c, 5a and 5b to include the details of the particular ratio we have measured.

7. The number of independent experiments that were quantified were not shown. Does the standard error in the bar graph represent multiple independent experiments or are these replicate lanes?

The standard errors in the bar graphs represent multiple independent experiments. The relevant n numbers have now been added to each Figure Legend.

8. Figure 3c: The unfractionated HA blot shows less zDHHC5 236/237AA than WT in the lysate. Can this account for the reduced amount of PLM seen in the co-IP vs. a reduction in the interaction?

We agree that there appears to be a little less of the 236/237AA mutant compared to wild type zDHHC5 in the unfractionated samples. However, the amount of PLM pulled-down in this experiment was normalised to the amount of recombinant zDHHC5 captured by the

HA beads. By expressing our data as a ratio, any slight differences in the expression of the wild-type and mutant constructs used had no impact on the interaction we were measuring. We have relabelled the y-axis on the bar chart to make this clear.

9. The specificity of co-IPs of integral membrane proteins is a concern. To argue that Figure 3f confirms is the only zDHHC protein to interact with zDHHC5, the authors should overexpress another DHHC protein and show that it doesn't pulldown zDHHC5. zDHHC2 is a reasonable candidate as it shows some plasma membrane staining when overexpressed. Alternatively, Figure 3f could be removed from the manuscript.

We have removed Figure 3f as suggested by the reviewer.

10. Figure 4a is not very convincing. The signal in the P boxes is weak and confidence is not increased when the gel is cropped to a single band.

We apologise for the inclusion of a poor-quality blot in our initial submission, and have replaced it here with a better one.

11. Figure 4c – the bands are continuous, raising questions about the quality of the data used for quantitation.

A further experimental repeat has been performed and a better-quality representative blot included in the revised version of the manuscript.

12. The TAT peptide experiment needs an additional control to show that treatment of the cells with a TAT peptide is not impacting PLM palmitoylation. One suggestion is to use a scrambled NBSP sequence, although there might be better controls described in the TAT peptide literature. Quantitation of the Ras palmitoylation should also be presented. The same issues described earlier (items 6 and 7) are particularly important here as this experiment is the punchline of the paper.

We have extensively revised Figure 5 and now present data generated using a stearate tagged version of the disruptor peptide along with a scrambled control. Quantification of the impact of this peptide on the palmitoylation of PLM and H-Ras in an engineered cell line is presented in Figure 5a. We are also now in a position to show that the stearate tagged peptide can specifically block PLM palmitoylation in ventricular myocytes too (Figure 5b).

13. Experimental Procedures: zDHHC5 peptide array – what are the conditions for solubilization of the rat ventricular lysates? What detergent was used and how was

insoluble material removed? Incomplete solubilization of membrane proteins can result in small membrane fragments that are only pelleted at high g force.

As part of her doctorate studies, FP performed a series of solubilisation experiments with 3 different detergents (C12/E10, Fos-Choline 12 and CHAPS) at different concentrations of detergent relative to a fixed quantity of myocyte material. Solubilisation was performed at 4°C for 1h, and the insoluble material removed by centrifugation at 20,000g. The abundance of Na pump in each detergent solubilised supernatant was determined by semi-quantitative Western Blotting, and a graph of Na pump extracted v detergent concentration plotted for each detergent. This enabled us to experimentally determine the optimal detergent concentration for cell permeabilization. These results are available within the Ph.D. thesis of Fiona Plain (University of Dundee, 2018). Following completion of this work, we used 2mg/ml C12E10 to prepare the rat ventricular lysates. This detergent was chosen as we have used it previously and successfully with the PLM / Na pump complex (relevant PMIDs – 14597563, 15621037, 17157829, 17283221, 18065526, 19339511, 21849407, 21868384, 23532852, 23612119, 25422474, 30165515).

Reviewer #2

1. Can the authors use crosslinking or a mutagenesis-based approach to nail down the residues that are important in the enzyme -substrate interaction, i.e. between the amphipathic helix and the Na pump?

We have now shown that intracellular loop 3 of the sodium pump alpha sub-unit interacts with zDHHC5 (Figure 2c, d). At this stage, however, we cannot say whether the interaction of zDHHC5 with this region of the pump alpha subunit is direct or via an adapter. During the process of resubmitting our manuscript, there have been two high-profile publications (Woodley, K.T., and Collins, M.O. (2019) S-acylated Golga7b stabilises DHHC5 at the plasma membrane to regulate cell adhesion. EMBO Rep., e47472; Ko, P.J. et al. (2019) A zDHHC5-GOLGA7 protein acyltransferase complex promotes non-apoptotic cell death. Cell Chem. Biol. 26, 1716-24) showing that the region of zDHHC5 we have identified as necessary for contacting the pump is also involved in binding GOLGA7b, a homolog of GCP16 which is part of the zDHHC9 enzyme complex. At this stage, despite considerable effort on our behalf, we have no evidence of a role for either Golga7 or Golga7b being the missing adaptor protein between zDHHC5 and the Na-pump alpha sub-unit. Indeed, when we conducted a full proteomic analysis of the proteins interacting with NBSP in cardiac tissue neither Golga7 nor Golga7b or any other related proteins were found to bind to it. Clearly, the NBSP region that we, and others, have identified is vitally important for zDHHC5 function, the molecular details of which have yet to be fully understood. Although this is of interest to us, we do not think that identification of (the) zDHHC5 adapter protein(s) in the heart (if indeed it/they exist) is a pre-requisite for publication of this manuscript as its central messages are that substrate recruitment to a zDHHC-PAT can be regulated by post-translational modifications, and that the ability of a zDHHC-PAT to palmitoylate one substrate can be selectively targeted pharmacologically by interfering with substrate recruitment. This said, we have modified our Discussion to include mention of Golga7/7b in the context of proteins that bind to the NBSP region of zDHHC5.

2) In the experiments with zDHHC20 mediated palmitoylation of zDHHC5, a control experiment with a catalytically inactive mutant, such as zDHHS20 would better confirm that the catalytic apparatus of zDHHC20 is indeed responsible for palmitoylating zDHHC5.

This experiment has been done as requested. Please see Figure 3g.

Reviewer #3

#1. In Figure 1A, the peptides used to fish the Na-pump are in fact pretty short, suggesting that the interaction of full-length zDHHC5 with the Na-pump is not depending on a complicate three dimensional fold. From this analysis, author pointed a particular alpha helix from the zDHHC5 enzyme covered by the sequence of the three positive peptides (Figure 1C). Additionally, the author indicate in the Discussion section that “In contrast to the relationship between some zDHHC-PATs and their substrates, the interaction between enzyme and substrate was sufficiently tight to enable purification of the Na-pump in complex with zDHHC5”. Therefore, could it be considered a reciprocal experiment with peptides obtained from the Na-pump that should be used to fish zDHHC5? I strongly suggest the author to consider such an experiment. Mapping the interaction patch of zDHHC5 on the Na-pump may indeed help greatly in the understanding of the mechanism of binding to and releasing from the Na-pump. From the available structure of the Na,K-ATPase/B-subunit/PLM-like protein (as shown in fact in the figure 7 of Tulloch et al. (2011) and in the figure 4C of Howie et al. (2014)), and considering the figure 1E of the present Ms, a zone on the Na-pump should be easily defined to help in the designing and choosing of the Na-pump peptides to be tested.

We thank the reviewer for these suggestions. We have worked on this point extensively and taken many different approaches to try and precisely map the site of interaction. We can now show that the third intracellular loop of the pump interacts with zDHHC5 (Figure 2c, d). However, we cannot yet unequivocally say whether or not an adapter protein is involved (Reviewer 2, point 1).

#2. The interaction between the Na-pump complex and zDHHC5 seems to be strong. The structure of the Na,K-ATPase/ β -subunit/PLM-like protein complex have been resolved in different conformations since the ten past years. The structure of zDDHC20 have been resolved last year, do the author consider the opportunity to decipher the structure of a Na,K-ATPase/B-subunit/PLM-like/zDDHC5 supercomplex? This proposal is only a recommendation and not a necessity for the present Ms. If zDHHC5 and zDDHC20 share a high homology, purification of decent amount of zDHHC5 can be maybe considered.

We have not addressed this comment in accordance with the Editor's instructions.

#3. A large set of techniques have been used to study protein-protein interaction in this paper. A very close parent of the Na-Pump (P20020, PMCA1) and close homologues of its accessory proteins (O75787, P54709) have been detected (Supplementary tables 1 and 2). Surprisingly, authors do not mention this result in the Discussion section. Does palmitoylation have been reported for these proteins to date?

Within Supplementary Table 2 we have provided an analysis of the overlap between our BirA datasets and SwissPalm, which is a catalogue of all palmitoylated proteins identified to date. PMCA1 and Na pump beta3 subunit (P54709) have both been found in proteomic screens of palmitoylated proteins. Given the sensitivity of mass spec approaches, however, experience has taught us to err on the side of caution when interpreting proteomic lists. Na pump beta3, for example, has no intracellular or transmembrane cysteine residues so it can't be palmitoylated. The problem with any proteomic approach (including proximity biotinylation) is that interesting 'hits' require independent verification to determine whether or not the screening results have biological meaning. We have not followed up any of the 'hits' the reviewer mentions, so we do not propose drawing attention to them in the discussion.

#4. A paper in 2014 (Montigny et al., doi: 10.1074/jbc.M114.590307) and a review in 2016 (Barbot et al., doi: 10.1007/978-3-319-24780-9_10) reported the palmitoylation of Sarcolipin (SLN), an accessory regulatory peptide of the sarco-endoplasmic reticulum Ca²⁺-ATPase (SERCA). SERCA is a very close parent of the Na-Pump. Therefore, these two references may be mentioned in the introduction.

Thank you for this suggestion. We have now included these references within the Introduction as requested (pages 4 & 5).

#5. Considering structural data, SLN binds to SERCA in a groove very close to the binding site of PLM on the Na-Pump. Surprisingly, PLM Cys40 and 42 are located at the membrane interface whereas SLN Cys9 is located deeply in the membrane (Winther et al. (2015), Toyoshima et al. (2015) and Barbot et al. (2016)). Do you think that the three pumps and their subunits (i.e. SERCA/SLN, Na-Pump/PLM and PMCA1) may share the same mechanism of palmitoylation?

We thank the reviewer for these comments. It is conceivable that the mechanism of PLM palmitoylation we present here is also applicable to SERCA/SLN and PMCA1. However, as we have no experimental data to suggest that this is indeed the case we are reluctant to speculate on this possibility. Furthermore, we are concerned that an extended discussion of molecular pumps and their possible regulation by palmitoylation will take away from the paper's main message – that a protein's palmitoylation status can be selectively manipulated by altering its recruitment to a cognate zDHHC-PAT. Therefore, we do not think it is appropriate to address these points, valid as they are, within the revised manuscript.

#6. As mentioned in comment #3, a large set of very recent techniques have been used in this study to look at protein to protein interactions which is a very good point. However, as these techniques are pretty recent and for now not really common, I suggest to clearly add a few words about the principle of each of them (in the Materials and Methods section or when describing the figure in the Result section) to help readers.

Text describing the proximity biotinylation approach has been included in the revised version of the manuscript.

#7. Several figures legends contain descriptions or conclusions that are paraphrasing the text without giving key informations on the way the experiments have been performed. These informations are also not available from the Methods section. Please avoid adding conclusions to figure legends. Replace them by a detailed description of the experiment (e.g. sample content, amount of protein loaded (if the same for each lane, it should be indicated for comparison of the level of expression in the UF fractions), detection system used, nature of the tags, nature of the antibody for co-immunoprecipitation, antibody used for detection). These parameters are absolutely necessary for the understanding of the experiment by the reader, to let them make their own analysis and conclusions. Conclusions of the authors should take place only in the result section.

All Figure Legends have been amended as requested.

#8. "Mutation of cysteine 245 was essentially without effect, suggesting that this residue is not quantitatively palmitoylated." may be replaced by "Mutation of cysteine 245 was without effect, suggesting that this residue is not palmitoylated"

Thank you for this suggestion. We have modified the text accordingly (page 8).

#9. "suggesting that either C-236 or -237 is palmitoylated (Fig. 3b)." may be replaced by "suggesting that C-236 and/or -237 are/is palmitoylated" after deduction from the effect of the C245A mutation. Why this mutation seems to trigger multiple palmitoylation of C236 and 237?

The acyl-PEG experiment in Figure 3b shows that either C236 or C237 is palmitoylated but not both. Wild type zDHHC5 is triply palmitoylated. zDHHS5 is double palmitoylated. zDHHS5 C245A is also doubly palmitoylated. zDHHS5 C236/7AA is singly palmitoylated. All together, this means that either C236 or C237 is palmitoylated. This agrees with our acyl-RAC data in Figure 3a which shows residual, identical capture of zDHHS5 C236/7AA and zDHHS5 C236/7/45AAA. These results imply that there must be an additional palmitoylation site located outside of the zDHHC5 C tail. This has been observed for other

zDHHC enzymes (e.g. J. Biol. Chem. 290, 29259-29269). We have clarified these points in our description of Figure 3 on pages 8 & 9.

#10. “suggesting that the presence of (a) membrane anchor(s) in this part of zDHHC5 promotes interaction with the Na-pump.” The difference between WT and mutant is not strong. Maybe "promotes" could be replaced by "contributes moderately to" to make the statement more elusive. Did authors check the level of expression of the WT and mutated zDHHC5. If the level of expression of the mutated version is significantly lower than the WT, the amount of Na-pump detected will be also lower, isn't it?

We have tempered our conclusion (page 9) as the reviewer suggests. With respect to the relative expression of WT and mutated zDHHC5 in Figure 3c we direct the reviewer to our response to point 8 from reviewer 1.

#11. “Furthermore, overexpression of HA-tagged zDHHC20 caused both increased zDHHS5...” I disagree as the level of expression decrease dramatically as observed from the UF loads? why? The level of expression of the mutant 236/7AA is very low compared to the WT and it seems to be not affected by overexpression of zDHHC20? “...palmitoylation (Fig. 3g) and increased co-immunoprecipitation of zDHHC5 with the Na-pump in HEK cells (Fig. 3h).” Authors initially claim that zDHHC5 is not interacting directly with PLM. Here, they would like to look at a direct interaction of zDHHC5 with the Na-Pump but they show western-blot from detection of PLM, assuming that Na-pump is also present. Why not using an antibody directed against the Na-Pump to have a direct detection?

The reviewer is correct that in the representative Western Blot presented in Figure 3f (formerly Figure 3g in the first version of the manuscript) the expression of zDHHS5 is different in the presence and absence of zDHHC20. However, when assessing palmitoylation of zDHHS5 (and indeed any other protein) we determine the abundance of the palmitoylated population relative to the total abundance of the protein of interest in the original extract (i.e. Palm / UF). This point has now been clarified in all Figures where palmitoylation data is presented. By measuring a protein's palmitoylation status in this way it is possible to obtain reproducible results even when we observe day-to-day and well-to-well differences in protein expression.

With respect to Figure 3h, we present co-purification of a Na-pump subunit as an index of pump association with zDHHC5.

#12. “the zDHHC5 C tail is highly disordered”. I do not find any structural data in ref 13 supporting this statement. I suggest writing “is predicted to be disordered” if some

prediction has been done, either the program or the method used should be mentioned.

We accept the reviewer's comment and have modified the manuscript accordingly (page 10).

#13. "the OGA inhibitor GlcNAc-statin (GNS) increased both PLM palmitoylation and co-immunoprecipitation between zDHHC5 and the Na-pump in cells transfected with wild-type zDHHC5 but not a S241A mutant (Fig. 4b, c). Together, this indicates that GlcNAcylation of zDHHC5 S241 promotes PLM palmitoylation through..." It is of course an indirect proof as zDHHC5 has seemingly a lot of different substrates in the cell as strongly suggested by pulldown assays done by the authors (Supplementary data 1 and 2). Inhibition of its GlcNAcylation may interfere with a lot of other processes and signalling pathways in the cell that may affect indirectly the Na-pump and its accessory subunit (e.g. their level of expression). To conclude, authors need to assess precisely the gain or loss of function of the Na-pump following overexpression of WT or S241A zDHHC5. If not possible, authors should only indicate: "this suggests that..."

We accept the reviewer's argument and have modified the manuscript in accordance with their suggestion (page 11).

#14. In the Discussion section, authors wrote "We cannot, however, rule out the possibility that the zDHHC5 NBSP contains a binding site for an adaptor protein that interacts with the Na-pump a sub-unit (and potentially other zDHHC5 substrates too)". It sounds unlikely as nothing resembling to a putative "adaptor" was detected by MS from the different protein-protein interaction assay used during this study. In absence of any evidence, I suggest to remove this paragraph.

See our response to Reviewer 2, point 1.

#15. "A major finding of our work is the observation that covalent modification (both palmitoylation and GlcNAcylation) of residues proximal to a substrate-binding site on a zDHHC-PAT affected both the recruitment and palmitoylation of that substrate – a phenomenon that will likely apply to many other zDHHC-PATs and enzyme-substrate combinations". Authors might be more careful. Since a modest modulation of the level of expression of the zDHHC5 enzyme itself (resulting from its post-translational modification) may also have an impact on the level of palmitoylation of its target, even the interaction with this target is not affected. If authors do not have any idea of the level of expression of WT and mutated zDHHC5 in the different experiments, they cannot conclude (e.g. in Fig 3A, if the amount of protein loaded on the gel is the same in each lane, there is

significantly more zDHHS5 than zDHHC5 in the UF sample).

We agree with the reviewer that in some experiments there are differences in the level of expression of both zDHHC5 and zDHHS5. However we suggest that the key experiments that assess recruitment to and palmitoylation of PLM by zDHHC5 following either its palmitoylation (Figures 3c, 3h) or GlcNAcylation (Figures 4b, 4c) have been carefully controlled to ensure that we can conclude that post-translational modifications in or near NBSP influence the ability of the Na-pump to access its binding site on zDHHC5. This conclusion is indirectly supported by our successful use of NBSP-based disruptor peptides to modify PLM palmitoylation (Figure 5).

#16. “This raises the intriguing possibility that the activity of the Na-pump may be responsive to nutrient (glucose, amino acid, fatty acid and nucleotide) availability through dynamic changes in PLM palmitoylation as a consequence of zDHHC5 GlcNAcylation.”. This sentence should be more developed or removed. A few references may be added if authors would like to illustrate this idea. A lot of different groups have already published and demonstrate a link between diet or starvation with the expression and activity of different Na,K-ATPase in different tissues.

This section of the discussion has now been removed.

#17. “For example, a molecule that prevented recruitment of the Na-pump to zDHHC5 would block PLM palmitoylation increasing the pump’s activity, and would be useful for treating heart failure and cardiac hypertrophy.” As there is a high sequence homology between P-type cations transporters, authors have to consider possible side effects of such a molecule. Authors have indeed detected other P-type ATPases in their pull-down assay using zDHHC5 for phishing (see comments #3).

We agree that selectivity will be a challenge and have revised the text on page 14 to emphasise this point.

#18. Comments on figure contents:

We note that reviewer 3 has requested that we add lane numbers to several of the Figures containing Western Blot images. In reply, we would appreciate editorial guidance on this matter. In the revised manuscript we have not yet added any lane numbers as requested as we think that the Blots are sufficiently well annotated to make lane numbering redundant.

Figure 1:

Panel b, please add a few residue numbers to help the reader to localized the residues cited in the text.

Panel d, a caption containing the amino-acid sequence used to generate this figure, with the same colour code, should be added for sake of clarity. Where is localized the VLCSS sequence that is common to the three positive overlapping peptides in panel a? The figure legend should indicate the program used for modelling. S241 cited below should be also indicated.

Panel e, what does this structure correspond to? Is it a model of zDHHC5 based on the structure of DHHC20? Method use to generate this figure should be indicated. A few residues should be also annotated (e.g. first and last residue from the NBSP, residues from the catalytic site).

We welcome these suggestions and have addressed the reviewer's comments through modifying both Figure 1 and its Legend.

Figure 2:

Panel a, in figure legend, authors claim "A series of peptides covering the PLM C-tail successfully purified endogenous Na pump α sub-unit but not recombinant HA-tagged zDHHC5 from HEK cells indicating that zDHHC5 does not interact with PLM directly". I do not agree with this statement as the peptide used only covers a small part of the PLM sequence. As PLM and zDHHC5 are transmembrane proteins, interaction can occur in the membrane or on the exoplasmic side also. No peptides covering those two regions were tested.

Panel b, lane numbers should be added on the figure and in the text to help readers.

"Peptide purified" should be replaced by NBSP for sake of clarity. What was the sequence of the NBSP used here? It should be indicated in figure legend. Authors said that no PLM was fished out with NBSP in FC-12 (lane 8 starting from the left). But some PLM is in fact hardly visible (as in Triton, lane 6). As the interaction between the Na-pump and PLM do not survive to FC-12 treatment, this PLM may be fished by the NBSP, isn't it? Another fair technical reason of the presence of this band is that some sample came from the next well. To check this latter hypothesis, authors should obviously reload the same samples on a new gel but in another order. Text in the result section should be adapted.

Comments on Figure 2a: we draw the reviewer's attention to our 2014 publication which established that the interaction between zDHHC5 and the Na-pump occurs in the cytoplasm. This is the reason that we concentrated on peptides representing the intracellular region of PLM. We have amended the description of these results on page 6 to improve the clarity of this section.

Comments on Figure 2b: while we agree that there is a very small quantity of PLM in the NBSP pulldown samples using both Triton and FC-12, the signal is unequivocally and

significantly less than that in the C12E10 sample and can, therefore, be regarded as background.

Figure 3:

Panel a, please add lane numbers on figures and in the text as requested above. Authors should also indicate in the legend what is exactly detected with HA and with H-Ras. A few words should be added in figure legend to explain how bars were deduced from the blots. Panel g, add lane numbers as indicated above. Why the level of expression of zDHHS5 is dramatically decreased in the presence of HA-zDHHC20 as observed when comparing UF lanes? It seems that the level of expression is not taken in account when generating bars.

See our response to point 11 above. We have now clarified how we calculate palmitoylation levels and have relabelled the y-axis on all relevant bar charts as well as reworded the associated Figure Legends.

Figure 4:

Panel b and c - add lane numbers.

We request editorial guidance on this comment.

Figure 5:

Figure legend may indicate that the untreated sample corresponds to the DMSO negative control for sake of clarity.

Figure 5 has been completely revised and now uses a scrambled peptide as a control.

REVIEWERS' COMMENTS:

Reviewer #1 (Remarks to the Author):

The authors have responded to the critiques in a comprehensive manner. The manuscript is improved with respect to the clarity of the presentation and the additional data provided strengthen the conclusions.

Reviewer #3 (Remarks to the Author):

COMMSBIO-19-0327A

Fiona Plain et al. addressed most of the reviewer's concerns and follow several of their proposal. When not, the authors now provide solid arguments and additional references to strongly support their conclusions. Text and figures were significantly improved. Following reviewers comments, authors made many efforts to provide new results notably by showing that the third intracellular loop of the Na pump interacts directly with zDHHC5 (Figure 2c, d). However, authors may address a few minor points considering these new results for sake of clarity. The addition of a few words in the results and discussion sections perhaps accompanied by a supplementary figure may be enough.

1. Lines 148-150: Authors wrote "The Na-pump α sub-unit could only be co-immunoprecipitated with (phospho-S68)PLM from the C12E10 lysate suggesting that Triton, SDS and FC-12 had caused dissociation of the Na-pump oligomer into individual sub-units (Fig. 2b)." Triton X100 may not fully denature the alpha subunit while SDS and probably FC-12 fully unfold the protein (Cohen et al. doi: 10.1074/jbc.M414290200). Therefore, disruption of the interaction is not only a result of the dissociation of the complex but most probably the consequence of a denaturation of the protein. Thus, this particular experiment do not show that the zDHHC5 interacts only with the alpha subunit. It shows that zDHHC5 interacts with the folded complex. Only a crosslinking experiment coupled to mass spec analysis can undoubtedly demonstrate an interaction with one or the other of the subunits as suggested by reviewer 2 in the previous version. Authors can reformulate to moderate their statement and focus more on the next experiment with IL3 and IL4 peptides that is of sufficient quality to warrant such a finding.

2. Lines 153-156: While the authors now clearly demonstrate an interaction between the Na pump and zDHHC5, they should indicate to which regions of the Na pump IL3 and IL4 corresponds exactly. The Methods section mentioned nothing except that they expressed peptides as GST fusion proteins. It is of course not enough to indicate "intracellular loop" in the text. The cytosolic domain of the Na pump is very large and divided in three sub-domains. Considering the topology of the Na pump, the 3rd and 4th intracellular loops may corresponds to the short loops connecting transmembrane spans 6 to 7 and 8 to 9, respectively. Is it right? Depending on the size of these IL3 and IL4, the authors could add a scheme depicting the Na pump structure and highlighting the position of these peptides/regions in supplementary figures (and also PLM and its cysteine, see comment below).

3. Lines 162-163: Authors wrote "The Na-pump binding site on zDHHC5 is close to three cysteine residues located at positions 236, 237 and 245." The cysteine residues are not only close to but are part of the interaction site. Authors should reformulate for sake of clarity.

4. Line 249-254 and Figure 1 legend, panel e: Authors wrote "The NBSP is located a considerable distance from the enzyme active site". Now considering the location of the interaction between the Na-pump and zDHHC5, and the position of PLM in the Na pump complex, is this distance really

considerable? Considering the structure of the Na pump complex, this distance is maybe compatible with a direct interaction of zDHHc5 and PLM. PLM is in fact close to the NBSP site (the Na pump L6-7 loop as expected above from the topology?) and its cysteine is not so far as it is located in the cytosolic leaflet. Even it is not possible to rule out the existence of an adaptor protein, it is not appearing essential considering these new inputs. Authors should add a few words in the text to speak their minds and again should precise to which region of the Na pump corresponds IL3. The supplementary figure proposed above could also help here in supporting the results.

5. Figure 5:

5.a. Data provided in this new figure are very interesting. Authors used H-Ras palmitoylation as a convenient negative control. As mentioned, H-ras is a substrate of zDHHc9, not of zDHHc5. Is there any other known substrate of zDHHc5 that DP can affect, especially from the long list of potential interactors provided in Suppl. Tables? Use of such another control may strongly support a specific role of the DP peptide and rule out, at least on this model, possible unspecific interactions especially in a pharmacological context.

5.b. The scrambled peptide is clearly less efficient than the "native" peptide. Therefore, it indicates that the fold is very important, suggesting that authors should measure their statement at lines 148-150 as indicated above in the present review.

6. Line 260: a missing space "thatGolga7b"

7. Line 262-264: If an adaptor protein exists, it is not because the authors do not identify it here that it must be a protein of novel function as written. Authors should reformulate considering that their analysis is not exhaustive even it allows to identify a lot of putative partners and interactors.

8. Authors added number of replicates and p values for most of the experiment presented. The program and the test used to calculate p values should be indicated in each legend or as a short additional paragraph in the Methods section.

Reviewed by C. MONTIGNY

Reviewer #3 (Remarks to the Author):

COMMSBIO-19-0327A

1. Lines 148-150: Authors wrote "The Na-pump α sub-unit could only be co-immunoprecipitated with (phospho-S68)PLM from the C12E10 lysate suggesting that Triton, SDS and FC-12 had caused dissociation of the Na-pump oligomer into individual sub-units (Fig. 2b)." Triton X100 may not fully denature the alpha subunit while SDS and probably FC-12 fully unfold the protein (Cohen et al. doi: 10.1074/jbc.M414290200). Therefore, disruption of the interaction is not only a result of the dissociation of the complex but most probably the consequence of a denaturation of the protein. Thus, this particular experiment do not show that the zDHHC5 interacts only with the alpha subunit. It shows that zDHHC5 interacts with the folded complex. Only a crosslinking experiment coupled to mass spec analysis can undoubtedly demonstrate an interaction with one or the other of the subunits as suggested by reviewer 2 in the previous version. Authors can reformulate to moderate their statement and focus more on the next experiment with IL3 and IL4 peptides that is of sufficient quality to warrant such a finding.

Section has been reworded as requested.

2. Lines 153-156: While the authors now clearly demonstrate an interaction between the Na pump and zDHHC5, they should indicate to which regions of the Na pump IL3 and IL4 corresponds exactly. The Methods section mentioned nothing except that they expressed peptides as GST fusion proteins. It is of course not enough to indicate "intracellular loop" in the text. The cytosolic domain of the Na pump is very large and divided in three sub-domains. Considering the topology of the Na pump, the 3rd and 4th intracellular loops may corresponds to the short loops connecting transmembrane spans 6 to 7 and 8 to 9, respectively. Is it right? Depending on the size of these IL3 and IL4, the authors could add a scheme depicting the Na pump structure and highlighting the position of these peptides/regions in supplementary figures (and also PLM and its cysteine, see comment below).

The regions of the Na pump alpha sub-unit that we have called IL3 ($\alpha\alpha$ 339-772) and IL4 ($\alpha\alpha$ 824-843) have now been clearly detailed in both the Methods and Result sections. Inclusion of a Supplementary Figure was considered unnecessary.

3. Lines 162-163: Authors wrote "The Na-pump binding site on zDHHC5 is close to three cysteine residues located at positions 236, 237 and 245." The cysteine residues are not only close to but are part of the interaction site. Authors should reformulate for sake of clarity.

The text had been modified accordingly.

4. Line 249-254 and Figure 1 legend, panel e: Authors wrote "The NBSP is located a considerable distance from the enzyme active site". Now considering the location of the interaction between the Na-pump and zDHHC5, and the position of PLM in the Na pump complex, is this distance really considerable? Considering the structure of the Na pump complex, this distance is maybe compatible with a direct interaction of zDHHC5 and PLM. PLM is in fact close to the NBSP site (the Na pump L6-7 loop as expected above from the topology?) and its cysteine is not so far as it is located in the cytosolic leaflet. Even it is not possible to rule out the existence of an adaptor protein, it is not appearing essential considering these new inputs. Authors should add a few words in the text to speak their minds and again should precise to which region of the Na pump corresponds IL3. The supplementary figure proposed above could also help here in supporting the results.

This section of the discussion has been re-written to bring greater clarity.

5. Figure 5:

5.a. Data provided in this new figure are very interesting. Authors used H-Ras palmitoylation as a convenient negative control. As mentioned, H-ras is a substrate of zDHHC9, not of zDHHC5. Is there any other known substrate of zDHHC5 that DP can affect, especially from the long list of potential interactors provided in Suppl. Tables? Use of such another control may strongly support a specific role of the DP peptide and rule out, at least on this model, possible unspecific interactions especially in a pharmacological context.

5.b. The scrambled peptide is clearly less efficient than the "native" peptide. Therefore, it indicate that the fold is very important, suggesting that authors should measure their statement at lines 148-150 as indicated above in the present review.

The Disruptor Peptide Results Section has been modified to state that Cav3 palmitoylation in myocytes is

unaltered by the disruptor peptide, providing further evidence that the effect observed on PLM palmitoylation is indeed selective.

When discussing these results, we have stated that we can't rule out the possibility that the disruptor peptide also alters the palmitoylation status of other proteins that interact with the NBSP region of zDHHC5.

6. Line 260: a missing space "thatGolga7b"

7. Line 262-264: If an adaptor protein exists, it is not because the authors do not identify it here that it must be a protein of novel function as written. Authors should reformulate considering that their analysis is not exhaustive even it allows to identify a lot of putative partners and interactors.

This section of the discussion has been reworded to bring greater clarity.

8. Authors added number of replicates and p values for most of the experiment presented. The program and the test used to calculate p values should be indicated in each legend or as a short additional paragraph in the Methods section.

The p values were calculated using GraphPad Prism. This detail has now been included in the Methods Section.